# Metabolic heterogeneity of tissue-resident macrophages in homeostasis and during helminth infection

Graham A. Heieis[1] ✉, Thiago A. Patente[1], Luís Almeida ◉[1], Frank Vrieling[2], Tamar Tak[1], Georgia Perona-Wright ◉[3], Rick M. Maizels[3], Rinke Stienstra[2] & Bart Everts ◉[1] ✉

Tissue-resident macrophage populations constitute a mosaic of phenotypes, yet how their metabolic states link to the range of phenotypes and functions in vivo is still poorly defined. Here, using high-dimensional spectral flow cytometry, we observe distinct metabolic profiles between different organs and functionally link acetyl CoA carboxylase activity to efferocytotic capacity. Additionally, differences in metabolism are evident within populations from a specific site, corresponding to relative stages of macrophage maturity. Immune perturbation with intestinal helminth infection increases alternative activation and metabolic rewiring of monocyte-derived macrophage populations, while resident TIM4+ intestinal macrophages remain immunologically and metabolically hyporesponsive. Similar metabolic signatures in alternatively-activated macrophages are seen from different tissues using additional helminth models, but to different magnitudes, indicating further tissue-specific contributions to metabolic states. Thus, our high-dimensional, flow-based metabolic analyses indicates complex metabolic heterogeneity and dynamics of tissue-resident macrophage populations at homeostasis and during helminth infection.

Macrophages possess diverse abilities to support immune defense, tissue integrity and homeostatic processes across bodily systems. The importance of these cells is underscored by their ubiquitous presence in mammalian tissues from early embryonic development[1,2]. Embryonically seeded macrophages persist as long-lived inhabitants, maintaining numbers through local proliferation, in addition to increasing contributions from differentiated circulating monocytes throughout adulthood[2]. As every tissue contains a distinct cellular and metabolic niche, macrophage development and persisting function are likely determined by the site of residence[3].

In the absence of immune perturbation, the rate at which blood monocytes contribute to the macrophage pool differs according to tissue[4]. Peritoneal macrophages, for instance, are predominantly resident cells expressing high levels of F4/80 and TIM4, but lack MHCII. With aging, these macrophages are slowly supplanted by monocyte-derived cells via an MCHII^HiF4/80^Lo intermediary stage that is visible at steady-state[4,5]. In contrast to most tissues, the homeostatic turnover of intestinal macrophages is more rapid due to tonic interactions between an abundance of commensal microbes[4,6]. Monocyte differentiation in the gut has elegantly been described as a "monocyte waterfall", in which Ly6C^HiMHCII^− monocytes flow through a Ly6C^+MHCII^+ phase to become Ly6C^−MHCII^+ macrophages[6,7]. More recently, an embryonically seeded and long-lived TIM4+ population of macrophages has also been described in intestinal tissues[8], though little is known how they differ from MHCII+TIM4− cells.

[1]Department of Parasitology, Leiden University Medical Center, Albinusdreef 2, 2333 ZA Leiden, The Netherlands. [2]Nutrition, Metabolism and Genomics Group, Division of Human Nutrition and Health, Wageningen University, 6708WE Wageningen, The Netherlands. [3]School of Infection and Immunity, University of Glasgow, 120 University Place, G12 8TA Glasgow, UK. ✉e-mail: g.a.heieis@lumc.nl; b.everts@lumc.nl

Macrophage function is critically dependent on cellular metabolic changes that occur during stimulation, hence there is growing appreciation that nutrient availability within the tissue microenvironment is a determinant of macrophage residence and responsiveness[3,9]. At present however, macrophage metabolism is mainly recognized as a dichotomy between inflammatory and regulatory macrophages primarily using in vitro culture systems. Yet, the diversity of macrophages within tissues requires a more in-depth metabolic characterization of in vivo populations to encompass their true heterogeneity[9]. While several valuable tools are available for easy assessment of metabolism in vitro[10], the difficulty in isolating sufficient cell numbers, while maintaining confidence that cellular metabolism does not lose fidelity from processing, has hindered progress towards understanding macrophage metabolism in vivo. Metabolic flow- or mass-cytometry has recently been pioneered as a method to overcome this bottleneck, and provide metabolic data at the single-cell level for immune cells[11–13]. Here, we applied this principle to tissue macrophages, using spectral-flow cytometry.

Here we report dynamic metabolic states between immature monocyte-derived and resident macrophages, as well as marked metabolic diversity within and across peripheral organs. In addition, we observe site-specific changes during helminth infections and that the most metabolically responsive cells during infection are not resident cells, but instead predominantly arise from recently recruited monocytes. These findings emphasize the advantage of flow-based analysis of metabolic targets, and reveal unique metabolic properties between distinct tissue-macrophage populations in health and disease.

## Results

### in vitro-differentiated macrophages acquire divergent metabolic phenotypes

We made a selection of nutrient transporters and metabolic enzymes of core metabolic pathways based on previous studies, and of known relevance to macrophage function[11,12,14] (Fig. 1a). Analysis of mouse bone marrow-derived macrophages (BMDMs) (Supplementary Fig. 1a, b) treated with inflammatory stimuli (LPS + IFNγ), to induce a classically activated phenotype, demonstrated increased features of glycolysis (GLUT1, PKM) and the pentose-phosphate pathway (G6PD), as well as an altered tricarboxylic acid (TCA) cycle (SDHA) (Fig. 1b). This expression profile was consistent with metabolic flux data, which showed an increased extracellular acidification rate (ECAR), representative of glycolysis (Supplementary Fig. 1c). In contrast, IL-4-stimulated BMDMs with an alternatively-activated phenotype maintained high expression of lipid (CD36, CPT1A) and amino acid (CD98) transporters relative to LPS + IFNγ stimulated macrophages, in line with a high oxygen consumption rate (OCR) detected by Seahorse (Fig. 1b, Supplementary Fig. 1c). Interestingly, the rate-limiting enzyme for fatty-acid synthesis (ACC1) was also elevated in IL-4-stimulated BMDMs (Fig. 1b). Consistent with a role for this fatty-acid synthesis is these cells[15], blocking ACC function abrogated M2 polarization with no effect on M1 (Supplementary Fig. 1d).

Exploiting the cross-reactive nature of the antibodies targeting metabolic markers, we assessed the metabolic state of GM-CSF or M-CSF differentiated human monocyte-derived macrophages (hMDMs), which promote pro- and anti-inflammatory features in hMDMs, respectively[16] (Supplementary Fig. 1e). Several parallels were observed between BMDMs and hMDMs, including higher GLUT1, PKM, G6PD and SDHA expression in inflammatory GM-CSF compared to regulatory M-CSF hMDM (Fig. 1c). These changes were also supported by increased ECAR and OCR readings via Seahorse analysis (Supplementary Fig. 1f). Unlike mouse BMDMs, the largest increase between hMDM conditions was observed for ACC1 expression. It has been shown previously that GM-CSF triggers substantial metabolic changes in hMDMs, similar to what we show here, however, fatty-acid synthesis was not investigated[17]. Altogether, these data show that analysis of

metabolic targets detects expected differences in both human and mouse macrophage metabolism in vitro.

### Metabolic heterogeneity in macrophages across tissues

An advantage of spectral flow cytometry is the ability to define autofluorescence (AF) as a channel, allowing superior signal resolution. However, the emission of varying AF spectra can become an obstacle for spectral "unmixing" due to heterogenous populations or high AF intensity, as exhibited by digested tissues and macrophages. We developed a workflow to define AF spectra for optimal unmixing, which we successfully applied to a panel with up to 30 targets (excluding AF) (Supplementary Fig. 2).

As each organ harbors its own metabolic microenvironment, macrophages must adjust metabolically according to the site of residence. We therefore compared macrophages across seven tissues (Supplementary Fig. 3) to determine potential links between location and metabolism. Resident populations from all tested tissues clustered separately following dimensional reduction and principal component analysis of metabolic target expression (Fig. 2a, b), though more similarity was observed between the small and large intestines, as well as between splenic red pulp macrophages and liver Kupffer Cells, the latter consistent with their shared expression of transcription factors, such as SpiC[18,19]. In agreement with prior work[20], large intestinal macrophages displayed higher expression of metabolic proteins than their small intestinal counterparts (Fig. 2c, Supplementary Fig. 4a). Splenic and liver macrophages, despite their similarities, were distinguished from each other by elevated ACC1 and CPT1A levels in the Kupffer Cells (Fig. 2c), which may link to their specific need for lipid handling. Lipid storage and catabolism is also an important function of alveolar macrophages[21,22], and though they expressed CD36 highly, its expression was on par or lower than that of colonic, splenic, liver, and peritoneal macrophages (Fig. 2c). The same observation was true of CPT1A. Alveolar macrophages did, however, also express comparably low levels of glycolytic targets in line with their low glycolytic capacity[21]. In contrast, brain microglia displayed consistently low relative expression of all targets, with the exception of GLUT1, supporting the critical reliance on glucose for these cells[23] (Fig. 2c). Peritoneal macrophages had consistently high expression of all targets relative to other tissue macrophage populations, with the exception of CD98, and to lesser extent CD36, which was significantly higher in splenic and liver macrophages (Fig. 2c). Of note, peritoneal macrophages incubated at 37 °C alongside digesting tissues maintained comparable expression to the same samples stored immediately on ice, indicating tissue processing steps did not significantly impact analyzed metabolic parameters (Supplementary Fig. 4b). Comparing our data to published bulk RNA sequencing[24] of the same tissue macrophages showed similar expression patterns, although GLUT1 and CD36 showed clear discrepancies (Supplementary Fig. 5).

To further assess their metabolic divergence, we clustered tissue macrophages using phenograph according to only metabolic protein expression (Fig. 2d). The majority of tissues yielded one or two prominent clusters that were unique to that site. The presence of two clusters appeared to be a division of overall expression, correlating with relative cell size, rather than the presence or absence of particular targets (Supplementary Fig. 4c). Multiple smaller clusters were defined in the lung, one of which resembled interstitial macrophages (Supplementary Fig. 4d). In contrast, the small and large intestine had overlapping clusters, but also the greatest diversity. Despite the overlap of colonic and small intestinal macrophages, there was appreciable variability in the abundance of each cluster between them, with the most prominent large intestinal cluster (pg10) exhibiting higher metabolic target expression than the most prevalent small intestinal cluster (pg06) (Fig. 2e). Thus it appears that, with the exception of the intestines, macrophages within the same tissue/location, are largely homogenous regarding their metabolism.

## Efferocytosis of tissue macrophages requires fatty acid synthesis

The efferocytotic receptor TIM4 is a defining marker of resident peritoneal and liver macrophages, and is expressed at much higher levels in those cells than in intestinal macrophages (Fig. 2f). In contrast, TIM4 is absent in microglia and alveolar macrophages. We noticed that ACC1 levels were particularly high in TIM4+ macrophages, such as those of the PEC and liver (Fig. 2c, f). As fatty-acid synthesis has been linked to phagocytic processes[25], we hypothesized that efferocytosis may similarly require increased fatty-acid

synthesis to support membrane expansion for the engulfment of apoptotic cells. Accordingly, peritoneal macrophages efficiently phagocytosed dying cells ex vivo, whereas alveolar macrophages lacking TIM4 and possessing low ACC1 expression, did not take up apoptotic cells in the same timeframe (Fig. 2f, g). Moreover, pharmacological inhibition of ACC activity blocked efferocytosis by peritoneal macrophages, as well as in vitro by BMDM (Fig. 2h). Overall these data support the use of determining metabolic protein expression to identify mechanistic links between tissue-macrophage metabolism and function.

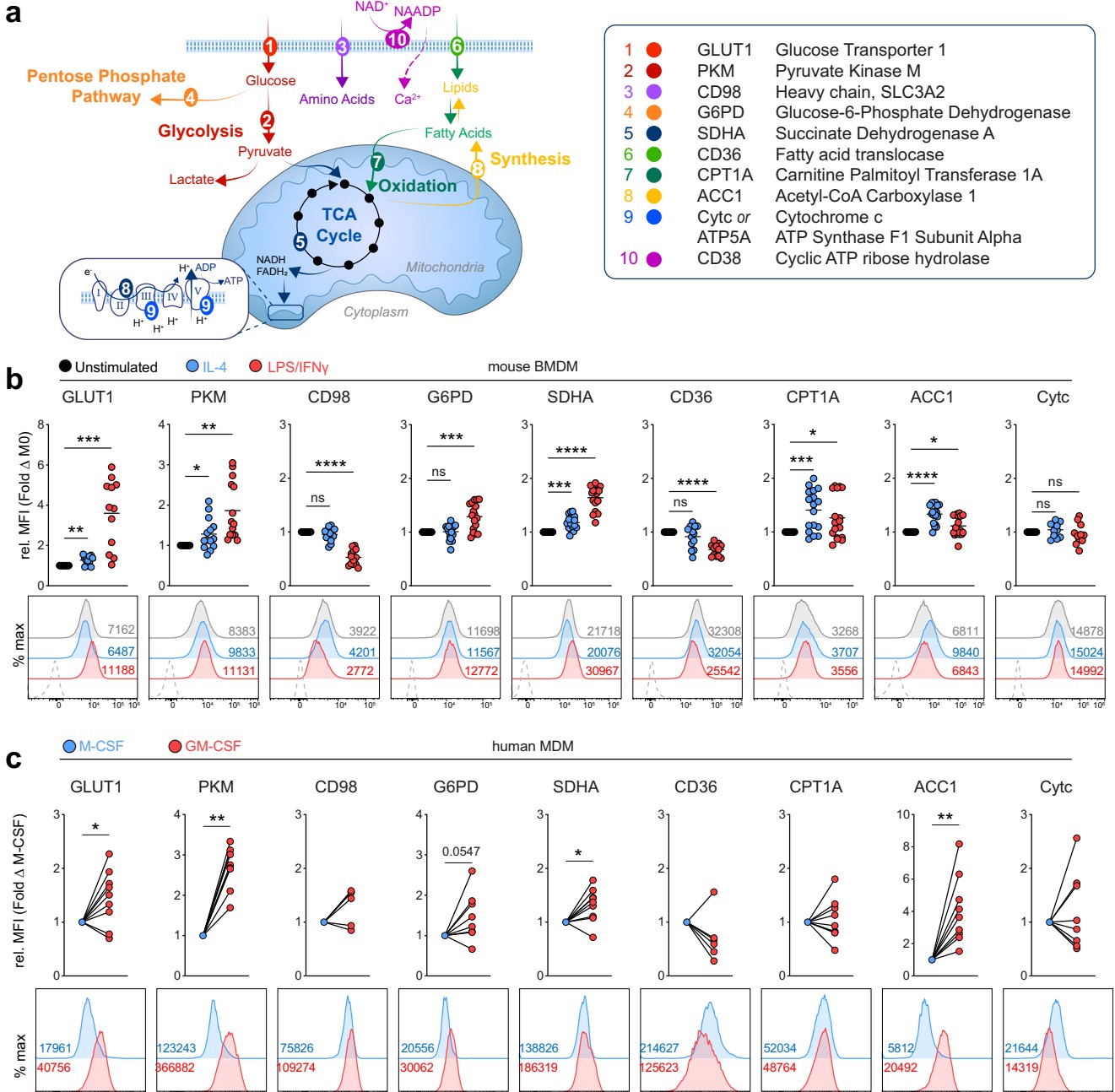

**Fig. 1 | Metabolic flow cytometry identifies distinct phenotypes between macrophages in vitro. a** Schematic of metabolic targets for flow-based analysis. **b** Relative change in geometric MFI of metabolic targets by stimulated BMDM relative to media control, and representative histograms below. Data points represent BMDM cultures from individual mice, pooled from 3–4 experiments ($n = 12$–16) with mean shown. **c** Fold change of metabolic target expression (gMFI)

in GM-CSF relative to M-CSF differentiated human macrophages, datapoints represent individual donors ($n = 6$–9). Statistics calculated with (**b**) RM one-way ANOVA with Geisser-Greenhouse correction and Dunnett Test to correct for multiple comparisons, or (**c**) Wilcoxon two-tailed matched-pairs signed rank test. ****$p < 0.0001$, ***$p < 0.001$, **$p < 0.01$, *$p < 0.05$. Source data are provided as a Source Data file.

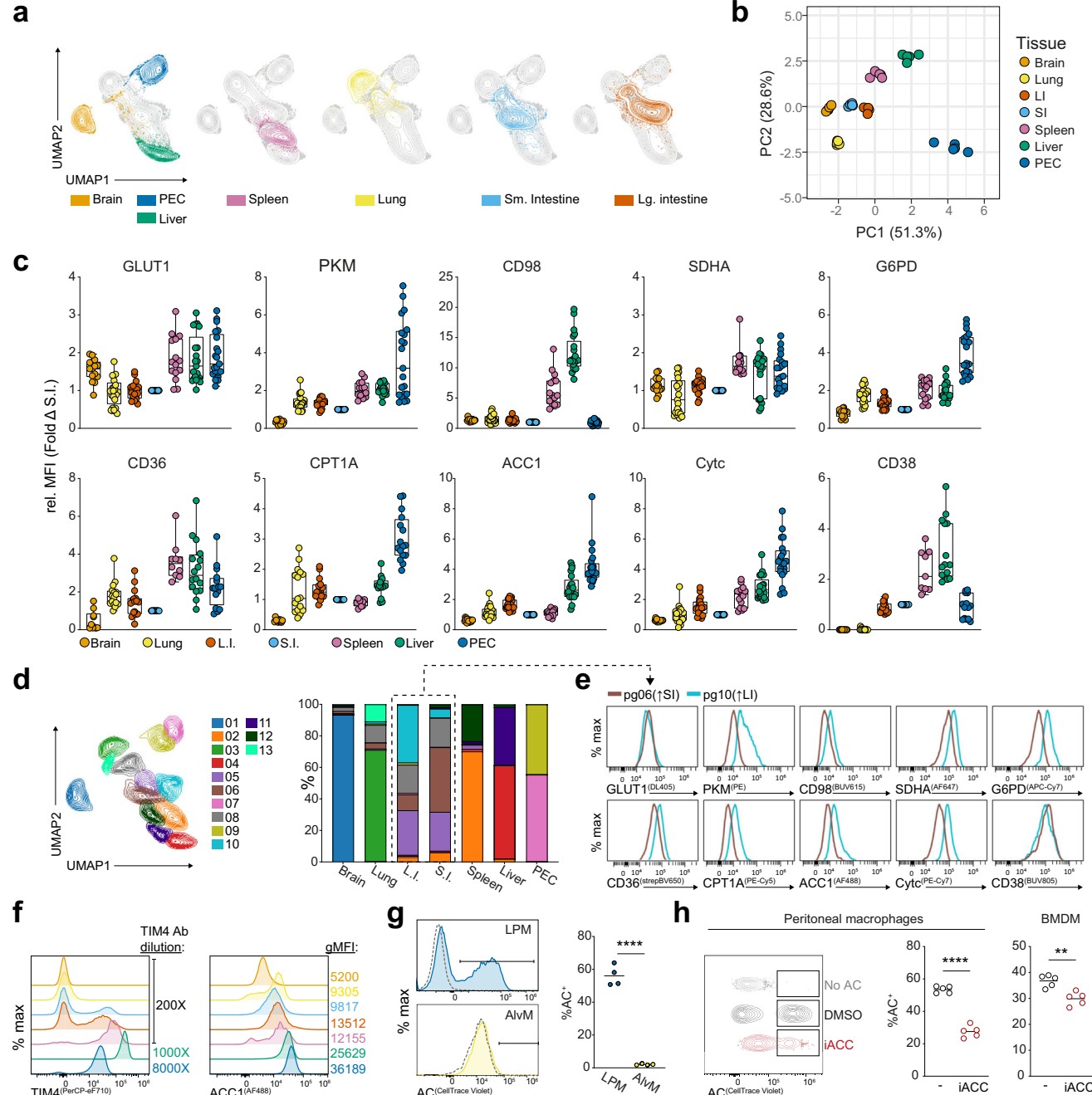

**Fig. 2 | Macrophages display unique metabolic properties across different tissues.** Brain (Lin⁻CD11b⁺), lung (Lin⁻CD64⁺), intestine (Lin⁻CD11b⁺CD64⁺MHCIIᴴⁱLy6C⁻), spleen (Lin⁻CD11bᴸᵒF4/80ᴴⁱ), liver (CD11bᴸᵒ/ᴹⁱᵈCD64⁺TIM4⁺), and peritoneal cavity (Lin⁻CD11b⁺) macrophages were gated in FlowJo and exported for analysis in OMIQ (**a**–**f**). **a** Uniform manifold approximation and projection (UMAP) analysis of all tissues using only metabolic targets. **b** Principal component (PC) analysis using metabolic target MFIs for tissue macrophages, done using the Clustvis web tool[59], representative of 3 independent experiments with n = 5. **c** Relative MFI of metabolic proteins normalized to the small intestine using brain microglia (CD11b⁺F4/80ᴸᵒCX₃CR1ᴴⁱ), alveolar macrophages (CD11b⁻SiglecF⁺), intestinal macrophages (Ly6C⁻MHCIIᴴⁱ), splenic red pulp macrophages, and resident peritoneal cavity macrophages (MHCII⁻F4/80ᴴⁱ), pooled from 3–5 independent experiments (n = 16–21, 25–75th percentile with median, min and max). **d** Phenograph (pg) clustering overlaid on UMAP dimensional reduction of tissue macrophages from (**a**) according to metabolic protein expression,

and frequency of clusters across tissues. **e** histogram showing intensity of metabolic targets in the most prominent clusters of the large (cluster 10) and small (cluster 6) intestines. **f** Histograms of TIM4 and ACC1 expression in tissue macrophages, shown according to ascending TIM4 expression. **g** Frequency of cell-trace violet⁺ cells after culturing either lung or peritoneal cavity macrophages with labeled apoptotic thymocytes for 1 h from 1 experiment; data points represent individual mice (n = 4) with mean shown. **h** Frequency of cell-trace violet⁺ peritoneal macrophages or BMDM incubated for 1 h with labeled apoptotic thymocytes or jurkat cells, respectively, after overnight pre-treatment with or without ACC inhibitor (60uM, CP 640,186), data points are from individual mice (n = 5), representative of 3 or 2 experiments, respectively. For (**c**) statistical tables can be viewed in Supplementary Tables 2–11, calculated using Kruskal–Wallis test with Dunn's multiple comparisons, data in (**g**) and (**h**) were compared using a two-tailed unpaired t test. ****p < 0.0001, ***p < 0.001, **p < 0.01, *p < 0.05. Source data are provided as a Source Data file.

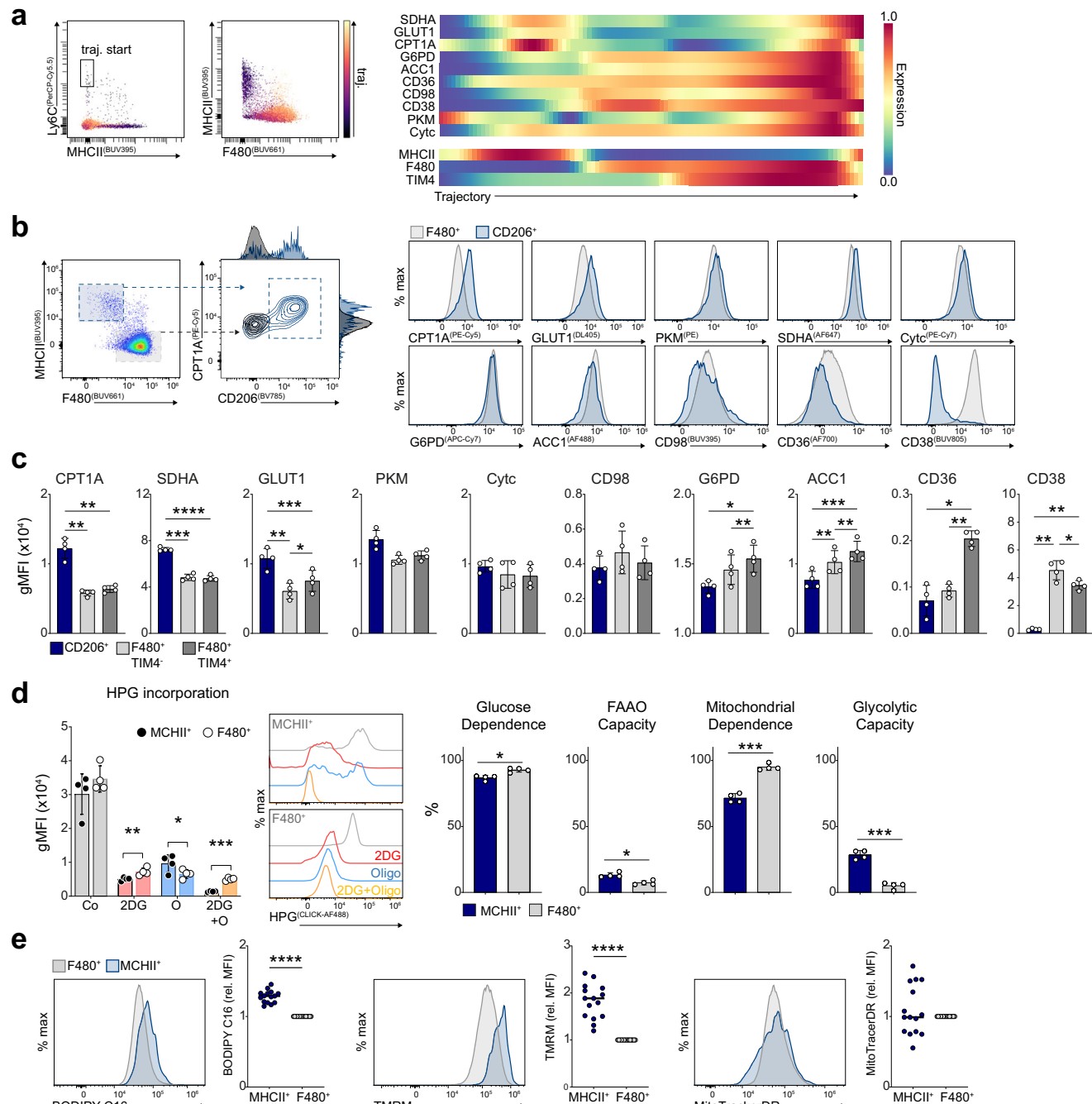

**Fig. 3 | Small and large peritoneal macrophages are metabolically distinct.**
**a** Wanderlust trajectory analysis of total CD11b⁺CD11c⁻ peritoneal exudate cells using immune marker (Ly6C, MHCII, F4/80, CX₃CR1, CD206, TIM4) and metabolic protein expression. **b** Manual gating of MHCII⁺CD206⁺CPT1A^Hi macrophages, and representative histograms comparing their metabolic marker expression relative to total F4/80^Hi macrophages. **c** gMFI of metabolic proteins in MHCII⁺CD206⁺, F4/80⁺TIM4⁻ and F4/80⁺TIM4⁺ populations. One of five representative experiments, compared using one-way RM ANOVA ($n = 4$, mean ± SD). **d** SCENITH analysis of cavity macrophages using HPG incorporation following incubation with 1uM

oligomycin (O), 100 mM 2-deoxy-d-glucose (2DG) or media controls (Co), one of two representative experiments shown with data compared using paired two-tailed $t$ test ($n = 4$, mean ± SD), FAAO = Fatty acid/amino acid oxidation capacity. **e** Long-chain fatty-acid (BODIPY C16) uptake, mitochondrial polarization (TMRM) and mitochondrial mass (MitoTrackerDR) of peritoneal macrophage populations, data pooled from 3 experiments of $n = 5$ mice and compared using Wilcoxon two-tailed matched-pairs test. ****$p < 0.0001$, ***$p < 0.001$, **$p < 0.01$, *$p < 0.05$. Source data are provided as a Source Data file.

## Dynamic metabolic states in peritoneal cavity macrophages

We then probed the intra-tissue metabolic heterogeneity of macrophages, starting with the peritoneal cavity. Application of wanderlust trajectory analysis to CD11b⁺CD11c⁻ cells using metabolic and immune markers predicted a trajectory from Ly6C⁺MHCII⁻ monocytes to F4/80⁺TIM4⁺ aligning with one of the proposed differentiation pathways for peritoneal macrophages[5] (Fig. 3a). Notably, expression of

metabolic targets changed dynamically over this trajectory. MHCII expression paralleled high GLUT1, SDHA and CPT1A expression, hallmarks of an highly catabolic metabolism, that was lost upon subsequent down-regulation of MHCII and acquisition of F4/80. Conversely, with progressively increasing F4/80 and TIM4 expression, markers of anabolic metabolism such as ACC1 and G6PD also increased. To more precisely associate metabolism with any particular

macrophage population we used dimensional reduction and Pheno-Graph clustering on total CD11b$^+$ cells (Supplementary Fig. 6a–c). Clustering according to immune and metabolic markers led to the separation of MHCII$^+$, F4/80$^+$TIM4$^-$ and F4/80$^+$TIM4$^+$ populations, and identified an MHCII$^+$CD206$^+$ cluster selectively enriched for GLUT1, CPT1A, and SDHA, that we were able to confirm by manual gating (Fig. 3b). In agreement with the trajectory analysis, these CD206$^+$ cells had comparatively less ACC1 and G6PD expression than both F4/80$^+$TIM4$^-$ and F4/80$^+$TIM4$^+$ macrophages (Fig. 3b, c). We also identified CD38, an ecto-NADase previously associated with inflammatory responses and aging[14], as a strict marker of mature F4/80$^+$ macrophages in homeostasis (Fig. 3c). Thus, trajectory analysis indicates a metabolic switch from anabolism to catabolism in monocyte-derived CD206$^+$ peritoneal macrophages.

We next questioned whether MHCII$^+$ macrophages also have different metabolic dependencies for ATP synthesis than F4/80$^+$ macrophages. To this end, we used an adaptation of the published SCENITH assay[26] that uses click-chemistry to measure the incorporation of a methionine analog, homopropargylglycine (HPG), into protein as a proximal readout for translation of which its rate is directly proportional to ATP availability (SCENITH-BONCAT)[27]. F4/80$^{Hi}$ macrophages exhibited a significantly increased dependence on mitochondrial ATP compared with MHCII$^{Hi}$ macrophages, in addition to a slight, but significant increase in glucose dependency (Fig. 3d). No differences were seen when comparing TIM4$^-$ and TIM4$^+$ cells (Supplementary Fig. 6d). Therefore, our data suggest an increase in catabolic pathways endow peritoneal MHCII$^+$ macrophages with a certain metabolic flexibility to maintain ATP synthesis, which could possibly serve as a means to mitigate metabolic stress within a new tissue environment.

Congruent with a more catabolic metabolism, MHCII$^+$CD206$^+$ macrophages also displayed greater fatty acid (BODIPY C16) uptake and mitochondrial membrane potential, despite having a similar mitochondrial mass (Fig. 3e). The increase in fatty acid uptake contradicted the lower surface CD36 that we observed on MHCII$^+$ relative to F4/80$^+$ macrophages, however, higher transcriptional expression of CD36 and of its transcriptional regulator PPARγ have been detected in MHCII$^+$ macrophages (Supplementary Fig. 6e, sourced from Immunological Genome Project Gene Skyline database), and we also observed an increase in surface CD36 on MHCII$^{Hi}$ macrophages after a 20 minute incubation at 37 °C (Supplementary Fig. 6f). Thus lower surface detection could be due to internalization of the transporter[28].

## Multiple metabolically distinct macrophage populations inhabit the intestinal lamina propria

Following our analysis across tissues, we pursued a further understanding of the marked diversity found amongst intestinal macrophages (Fig. 2d, e). Monocyte-to-macrophage conversion in the intestine is accompanied by global transcriptional changes[29]. Accordingly, we found that this transition paralleled significant changes in the protein expression of metabolic targets (Fig. 4a). Mature macrophages possessed progressively decreasing GLUT1 and ACC1 expression, but increased expression of all other targets, more in line with a transition towards catabolism. Interestingly, and similar to the cavity macrophages, CD38 seemed to strictly associate with mature macrophages. Despite these findings, mature macrophages showed a decrease in mitochondrial mass and membrane polarization relative to Ly6C$^+$ and intermediate monocytes (Fig. 4b). This was in direct contrast to the large intestine, where a similar transition was observed via met-flow (Supplementary Fig. 7a), and MHCII$^+$Ly6C$^-$ macrophages possessed significantly greater mitochondrial mass and potential over immature monocytes (Fig. 4c). It therefore appears that, in spite of acquiring enhanced metabolic machinery, mature macrophages in the small intestine poorly maintain their mitochondrial fitness following differentiation.

Further analysis of small intestine Ly6C$^-$MHCII$^+$ macrophages revealed multiple clusters based on immune and metabolic protein expression (Fig. 4d). In particular, three distinct TIM4$^+$ clusters were identified that could be segregated by CD11c and CD206 (Fig. 4d, Supplementary Fig. 7b). The CD11c$^+$TIM4$^+$ cluster had similar or higher expression of metabolic targets overall, with increases in ACC1, GLUT1, SDHA, G6PD, and CD98 over the CD206$^{+/l.o}$ clusters (Fig. 4e). However, the metabolic phenotype of the TIM4$^+$ clusters more closely resembled their matching TIM4$^-$ counterparts than each other (Fig. 4e), indicating metabolic phenotypes may be independent of ontogeny in the small intestine. Indeed, analysis of manually gated CD11c$^+$ and CD206$^+$ macrophages confirmed that metabolic phenotypes were more similar according to these markers, rather than the presence of TIM4 (Supplementary Fig. 7c, d). Interestingly, despite comparable metabolic marker expression between, TIM4$^+$ CD11c$^+$ and TIM4$^-$CD11c$^+$ the former showed a selective decrease in mitochondrial mass and membrane potential (Supplementary Fig. 7e).

Concurrent analysis of the large intestine exposed comparable CD11c$^+$ and CD206$^+$ macrophages within the TIM4$^+$ compartment (Fig. 4f). In contrast to the small intestine, however, the CD206$^+$ macrophages had similar or higher expression of metabolic targets over the CD11c$^+$ macrophages in the large intestine, which was matched by increased mitochondrial mass and potential (Fig. 4g, Supplementary Fig. 7f). The metabolic profiles of CD11c$^+$ and CD206$^+$ macrophages appeared to be a consequence of the tissue environment, as direct comparisons of these populations between both intestine sites showed large intestine CD206$^+$ macrophages had overall higher expression of proteins than matching cells from the small intestine (Fig. 4h). CD11c$^+$ macrophages on the other hand, possessed similar or higher expression in the small compared to large intestine (with exception of G6PD) (Fig. 4h). As the large intestine harbored a higher frequency of TIM4$^+$ and CD206$^+$ cells, these compounded differences explain the general increase in expression of metabolic targets in total MHCII$^+$ macrophages from the large intestine, over those from the small intestine (Supplementary Fig. 7g, Fig. 2c). Altogether, these data underscore a lack of homogeneity in TIM4$^+$ resident macrophages, and between macrophages of the small and large bowels.

We wondered if the metabolic differences observed between the large and small intestines also had any potential relationship with immune function. With this in mind, we noticed that small intestine macrophages had significantly elevated expression of the immuno-regulatory marker PD-L1, which appeared to be concomitantly expressed with CD11c (Fig. 4i). To further link PD-L1 to metabolism, we looked at the correlation between its expression and each metabolic target in both the small and large intestine (Supplementary Table 12). This uniquely identified a strong inverse correlation between PD-L1 and ACC1 (Fig. 4j). Although this conflicts with the elevated ACC1 observed in CD11c$^+$ compared to CD206$^+$ macrophage in the small intestine, further analysis showed highest ACC1 expression in a subset of PD-L1$^-$ cells (Fig. 4j), providing support of a possible counter-regulatory link between fatty acid synthesis and PD-L1 in macrophages residing in the intestines.

## Metabolic changes in intestinal helminth infection are restricted to recruited macrophages

Macrophage metabolism is commonly assessed using polarizing stimulations in vitro, yet how metabolic responses to these stimuli translate to tissue macrophages remains largely unanswered. Alternative activation in response to IL-4, for instance, causes metabolic reprogramming towards glycolysis[15], lipolysis[30] and glutaminolysis[31], but this has yet to be physiologically validated in vivo. *Heligmosomoides polygyrus* (Hp) is an enteric parasite that leads to alternative activation of macrophages in the small intestinal lamina propria as well as the peritoneal cavity, despite strict localization to the proximal small intestine. Hence, we characterized metabolic and polarization

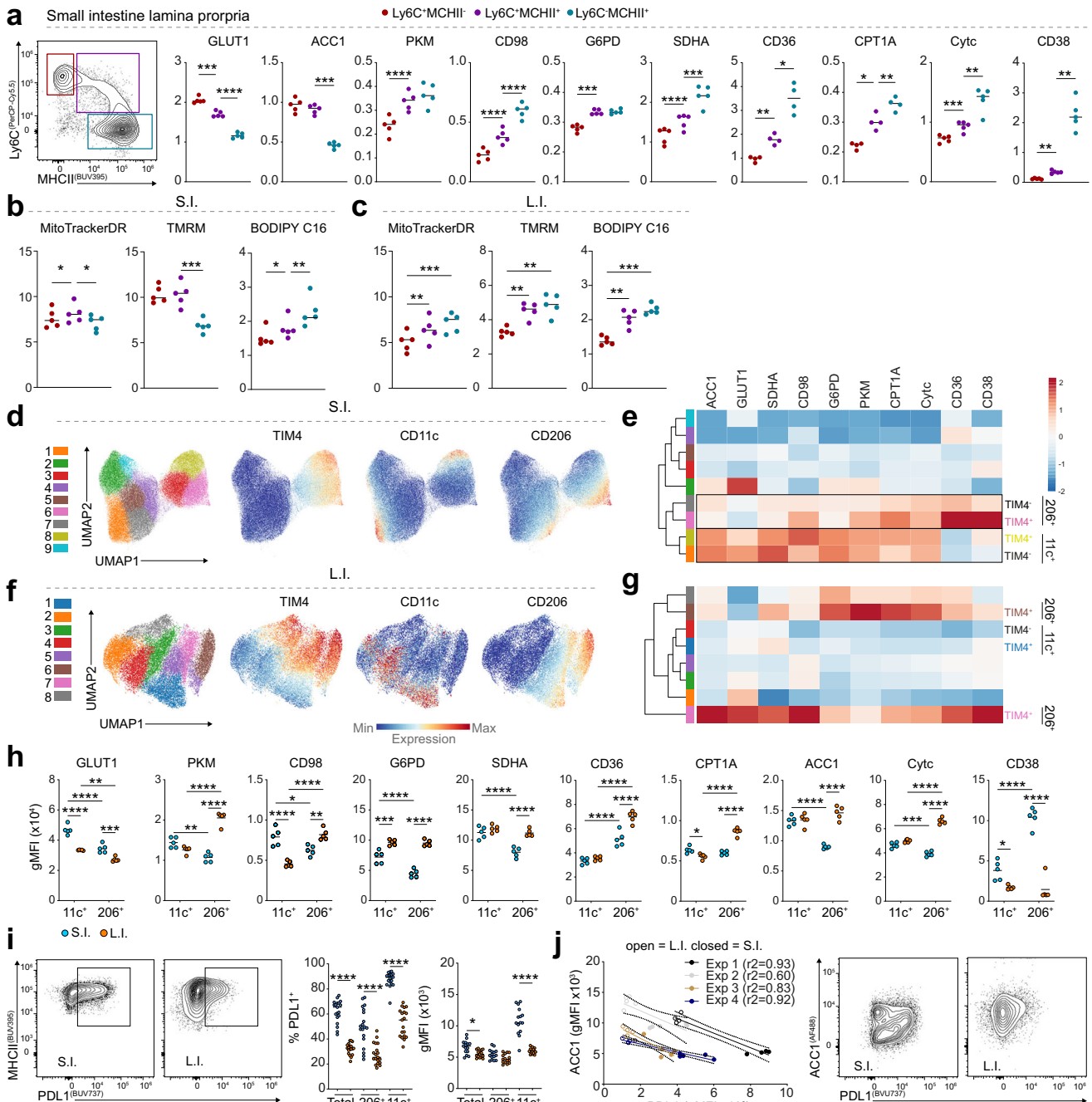

**Fig. 4 | Distinct immune phenotypes within intestinal TIM4+ macrophages linked to differing metabolic states. a** Gating for monocytes and intermediate or mature macrophages in the small intestine lamina propria, and corresponding gMFI values of metabolic targets. **b** TMRM, MitoTracker DeepRed and BODIPY C16 staining or uptake for monocytes/macrophages in the small and (**c**) large intestine. **d** UMAP plots and overlaid phenograph clusters generated using immune (MHCII, CD11c, F4/80, PDL1, CX3CR1, CD64, CD206, TIM4) and metabolic targets. **e** Heatmap with Euclidean clustering showing mean relative expression of metabolic targets across resulting phenograph clusters. **f**, **g** Matching analysis of the colonic lamina propria macrophages (corresponding to **d** and **e**). **h** Metabolic target gMFI comparing small and large intestine macrophages, according to CD11c or CD206 expression. **i** Representative PDL1 staining (left), frequencies of PDL1+ cells amongst total MHCII+ macrophages or within the CD206+/CD11c+ populations (middle), and corresponding gMFI of gated PDL1+ cells (right). (**j**) Correlation between ACC1 and PDL1 expression (gated on total MHCII+ macrophages) using a simple linear regression (dashed lines = 95% confidence interval), and representative staining of ACC1 vs PDL1 in the small and large intestine. Statistics done using either RM one-way ANOVA with Geisser-Greenhouse correction and Tukey test for multiple comparisons (**a**–**c**, **e**, **i**), two-way ANOVA and Tukey test for multiple comparisons (**h**, **i**). Representative of (**a**–**h**) or pooled from (**i**, **j**) four independent experiments. ****$p < 0.0001$, ***$p < 0.001$, **$p < 0.01$, *$p < 0.05$. Source data are provided as a Source Data file.

profiles of macrophages from the PEC and lamina propria (duodenum/jejunum) of naïve and infected mice (Fig. 5a, b).

Infection led to a global increase in metabolic enzyme/transporter expression in MHCII−F4/80Hi peritoneal macrophages (Fig. 5b). In contrast, Ly6C−MHCII+ macrophages from the infected intestine displayed more restricted increases, limited to glycolytic proteins and ACC1 (Fig. 5b). The frequency of TIM4+ cavity macrophages remained comparable between naïve and infected mice, consistent with the expansion of resident macrophages in response to infection (Fig. 5c). Indeed, peritoneal macrophages increased metabolic marker

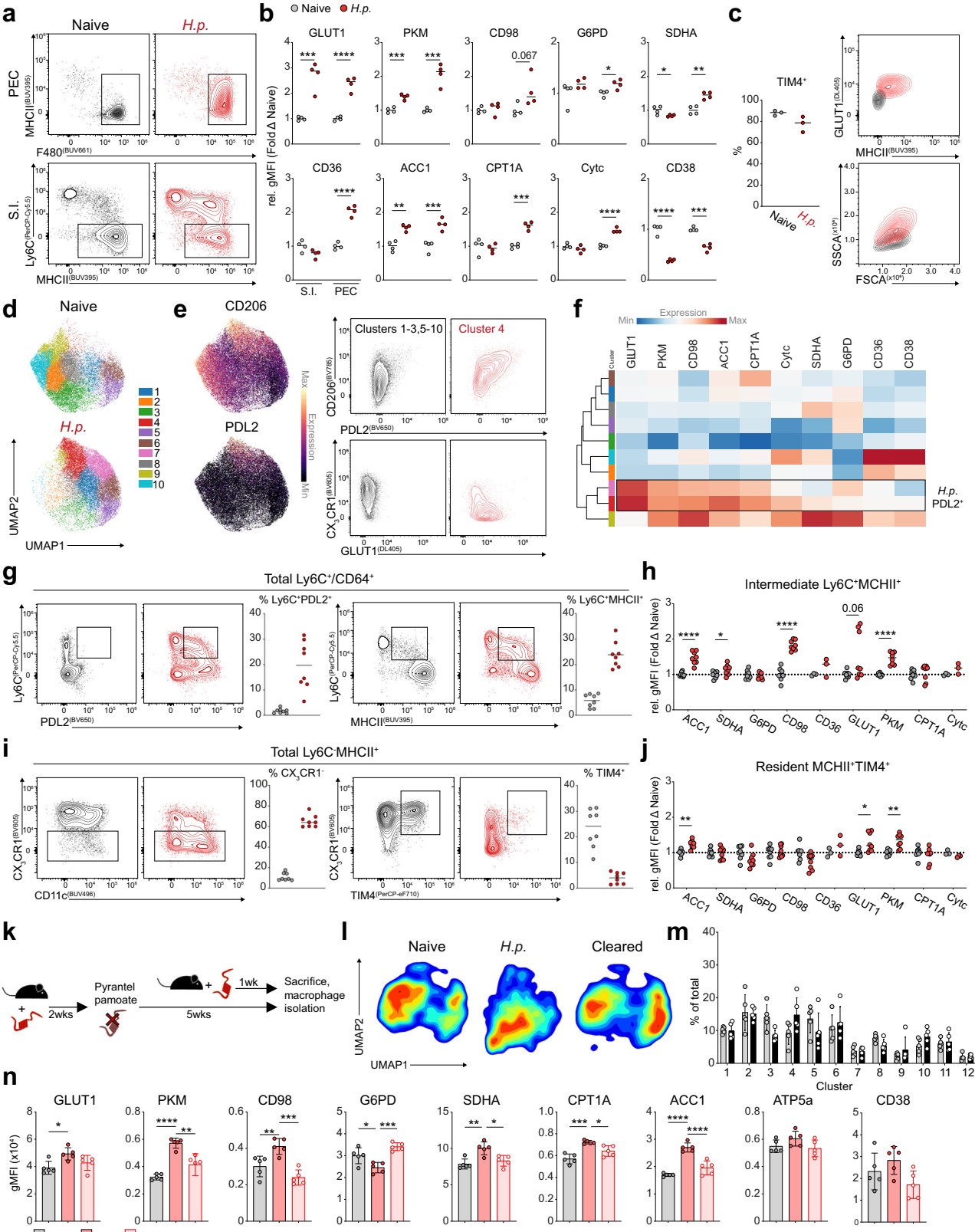

expression along with surface MHCII during infection, and displayed a larger, more granular phenotype (Fig. 5c).

Conversely, in the intestine, Hp infection led to a significant increase in monocyte recruitment and differentiation (Fig. 5a, g). After clustering intestinal MHCII⁺Ly6C⁻ macrophages using their expression of metabolic targets, we found one metabolically "high" cluster

belonging to the infected group (cluster 4) that we identified as alternatively activated macrophages according to their expression of PDL2 and CD206 (Fig. 5d–f). However, further analysis indicated that these cells are predominantly CX₃CR1⁻ consistent with their recent differentiation from immigrating monocytes[32] (Fig. 5e). Accordingly, amongst total macrophages, PDL2 expression was prevalent in Ly6C⁺

**Fig. 5 | Metabolically responsive, alternatively activated macrophages resemble monocyte-derived cells in the intestine during *H. polygyrus* infection.**
**a** Representative flow-plots of monocyte/macrophage populations from the peritoneal cavity (top) or intestine (bottom) from naïve or day 7 infected mice. **b** Relative gMFI of metabolic markers normalized to the mean expression in macrophages, as gated in (a), from the corresponding naïve tissue ($n = 4$/group) representative of 3 experiments. **c** Frequency of TIM4$^+$ macrophages in the naïve and infected peritoneal cavity, and representative plots of corresponding MHCII/GLUT1 expression and scatter profiles ($n = 3$/group) representative of 2 experiments. **d** Phenograph clustering overlaid on UMAP generated from metabolic target expression for Ly6C$^-$MHCII$^+$ intestinal macrophages. **e** Localized CD206 and PDL2 expression on UMAP and representative plots of PDL2$^+$ clusters from infected mice, compared to PDL2$^-$ clusters. **f** Heatmap showing mean metabolic expression for phenograph clusters. **g** Flow plots and graphs exhibiting Ly6C and PDL2 expression in the small intestine during infection, pooled from 2 experiments with $n = 8$ (**g–j**). **h** Fold change in marker expression on intermediate macrophages relative to the mean of the matching naïve population. **i** Representative CX3CR1, CD11c, and TIM4 staining and frequencies of total CX3CR1$^-$ cells, or TIM4$^+$ cells on MHCII$^+$ macrophages. **j** Fold change in marker expression on TIM4$^+$ macrophages relative to the mean of the matching naïve population. **k** Schematic for analyzing macrophage metabolism after helminth clearance. **l** Contour plots of UMAP generating from metabolic marker expression in MHCII$^+$ macrophages. **m** Frequencies of phenograph clusters for MHCII$^+$ macrophages from naïve (gray) or cleared (black) mice (mean ± SD). **n** Expression of metabolic targets in MHCII$^+$ intestinal macrophages from naïve, infected or cleared mice (mean ± SD). Data representative of, or pooled from, two experiments with $n = 2$–5 per group (**l–n**). Statistics done using either two-tailed unpaired $t$ test (**b, c, g–j, m**) or one-way ANOVA with Tukey test for multiple comparisons (**n**). ****$p < 0.0001$, ***$p < 0.001$, **$p < 0.01$, *$p < 0.05$. Source data are provided as a Source Data file.

intermediates (Fig. 5g). The presence of PDL2$^+$ cells in the intermediate macrophage population indicates alternatively activated macrophages during Hp infection could be of monocytic origin, and differences in ontogeny may confound metabolic readouts. As such, we observed significant metabolic changes in Ly6C$^+$MHCII$^+$ intermediate macrophages (Fig. 5g, h).

Within the Ly6C$^-$MHCII$^+$ population we further found near complete outgrowth of the TIM4$^+$ resident macrophages by CX$_3$CR1$^-$ macrophages (Fig. 5i), providing an additional indication that the majority of intestinal macrophages are replaced via rapid monocyte influx following Hp infection. To determine whether the metabolic changes we observed in total MHCII$^+$ cells were a result of responding monocyte-derived, and not resident macrophages, we compared the remaining TIM4$^+$ cells from infected intestines with those from naïve. In spite of the metabolic changes seen at the whole population level, TIM4$^+$ macrophages had strikingly similar expression between naïve and infected, although significant increases were still present for ACC1, GLUT1, and PKM (Fig. 5j). Moreover, no differences were observed in mitochondrial mass or polarization between TIM4$^+$ cells, yet these features were significantly increased in TIM4$^-$PDL2$^+$ macrophages (Supplementary Fig. 8a). TIM4$^+$ macrophages from the infected intestine also largely failed to acquire the alternative activation markers RELMα or PDL2 (Supplementary Fig. 8b), affirming that resident intestinal macrophages are metabolically and immunologically hyporesponsive.

The dominant role of macrophages in primary Hp infection is likely tissue repair, whereas they may contribute directly to resistance in challenge infection[33]. We asked whether the responding macrophages from first encounter remain metabolically "trained" after clearing the parasite, which could support their anti-parasitic function during second encounter. Five weeks post-treatment, the macrophage compartment of previously infected mice was comparable to mice that were left uninfected (Supplementary Fig. 8c). Expression of metabolic targets in intestinal macrophages also returned to pre-infection levels, such that naïve and cleared mice overlaid after dimensional reduction, and no differentially abundant clusters were found between them (Fig. 5k–m). Alternative activation markers similarly returned to baseline, as did metabolic markers in the peritoneal cavity macrophages (Supplementary Fig. 8d&e). Intestinal macrophages therefore undergo temporal changes in metabolism in response to helminth infection that do not persist long-term following clearance.

### Alternatively activated macrophages are highly metabolically active across tissues and infections

Considering the distinct metabolic phenotypes between tissues at steady-state (Fig. 2), we employed additional helminth infection models to determine similarities in alternatively activated macrophage metabolism at different sites. We first used chronic *Schistosoma*

*mansoni* (Sm) infection to look at the response of liver macrophages. Macrophages isolated from 16-week infected livers clustered completely separate to those harvested from naïve livers according to metflow targets (Fig. 6a). At this timepoint, infected mice had completely lost all TIM4$^+$ Kupffer Cells, while PDL2 expression was significantly elevated.(Fig. 6b, c). Approximately 40% of macrophages from infected livers were RELMα$^+$, all of which appeared to co-express PDL2 (Fig. 6c). We observed significant increases in expression of metabolic markers in RELMα$^+$ macrophages, whereas RELMα$^-$ macrophages remained similar to TIM4$^+$ Kupffer Cells (Fig. 6d). Yet, both RELMα$^-$ and RELMα$^+$ populations failed to acquire CD36, CD98 and CD38 expression to similar levels as macrophages from the naïve liver. As these markers are highly expressed in steady-state Kupffer Cells, it may reflect incomplete acquisition of a resident phenotype. An analogous phenomenon was identified in the lung of mice infected with *Nippostrongylus brasiliensis* (Nb). At 10 days post-infection, most Siglec F$^+$ alveolar macrophages were displaced by CD11b$^+$ macrophages, but both populations displayed evidence of a globally increased metabolism when compared to their corresponding naïve populations (Fig. 6e, f). The magnitudes of change were also similar in CD11b$^+$ and Siglec F$^+$ macrophages, although GLUT1 and PKM were greater in CD11b$^+$ cells, fitting with the hypothesis that alveolar macrophages have impaired glycolytic reprogramming in response to IL-4[21]. However, this is in part due to Siglec F$^+$ macrophages having higher expression of these targets prior to infection. Multiple states of alternative activation were found according to PDL2 and RELMα expression in both subsets, and in both cases the PDL2$^+$RELMα$^+$ cells exhibited the highest metabolic target expression, which was also seen in Hp infection (Fig. 6g, Supplemental Fig. 8f).

Lastly, to determine if we could identify a common metabolic phenotype linked to helminth-driven alternative activation, we compiled the relative expression of metabolic proteins in PDL2$^+$RELMα$^+$ macrophages to their naïve reciprocals (Fig. 6h). Ultimately, responding macrophages from each infection model displayed dramatic metabolic alterations. Increases in glucose (GLUT1, PKM), lipid (ACC1, CPT1A) and mitochondrial metabolism (Cytc/ATP5A) were characteristic across tissues, but to varying degrees. Our main findings from these data seem to highlight that the metabolic responses of macrophages to helminth infection are likely a consequence of both activating signals (i.e. IL-4) and the state of differentiation from monocytic origin.

## Discussion

Despite the widely recognized limitations surrounding in vitro experiments, a prolonged lag in pushing towards in vivo studies has hindered advances in macrophage immunometabolism. Flow-based analysis of metabolic enzyme expression has been demonstrated as a technique to investigate metabolic features of immune cell populations from blood[11,12] or tumors[13,34]. Here, we present the use of spectral

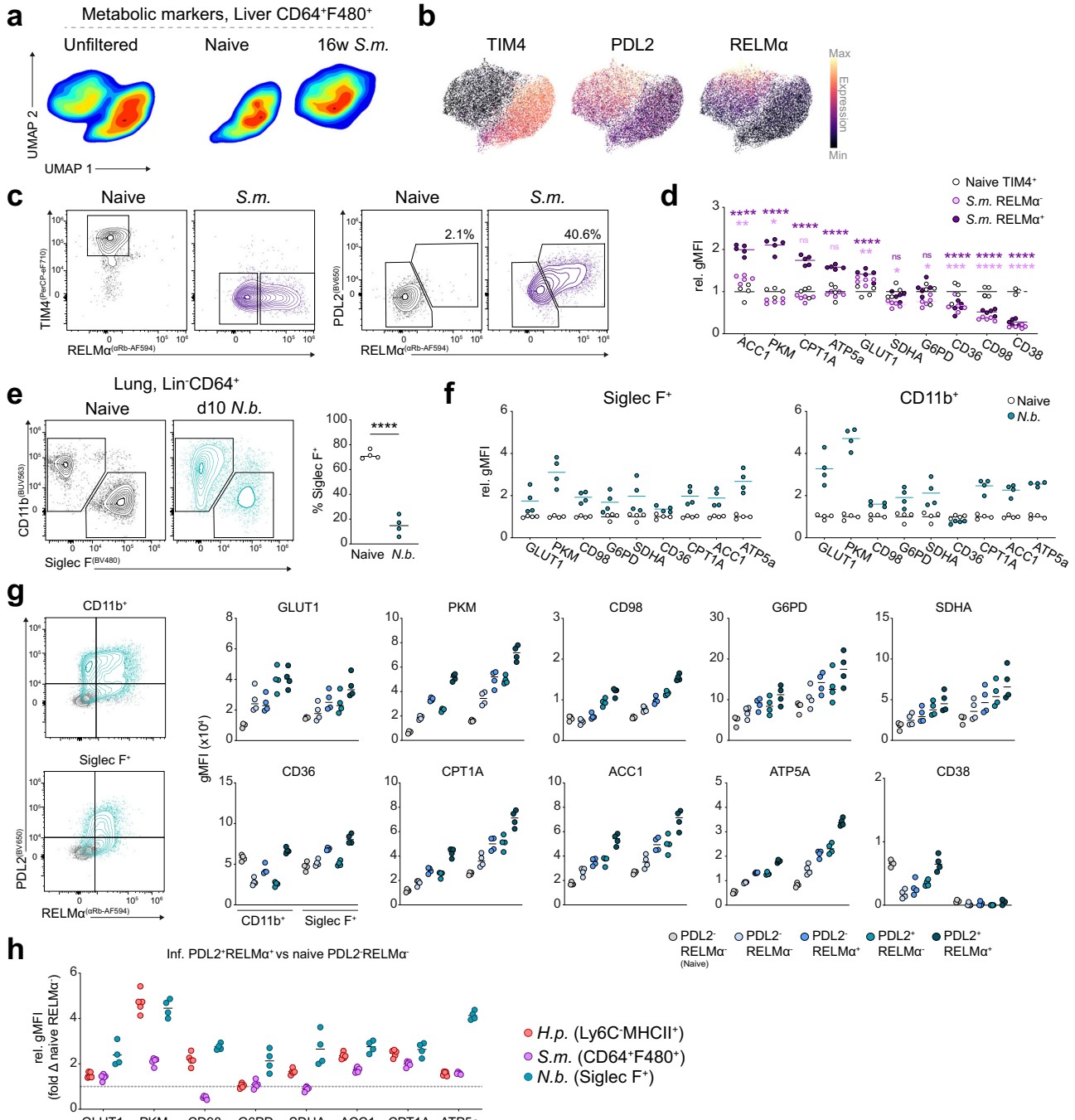

**Fig. 6 | Shared metabolic phenotypes in alternatively-activated macrophages from different models of helminth infection. a** UMAP of macrophages from the livers of naïve or 16 week *S. mansoni* infected livers and (**b**) overlay of TIM4, PDL2 and RELMα expression. **c** Representative gating for naïve resident or S.m induced alternatively activated macrophages. **d** relative expression of metabolic marker normalized to mean expression on naïve Kupffer cells, *n* = 4–6 mice per group shown as individual data points and mean. **e** Gating for lung macrophages in naïve and *N. brasiliensis* infected mice, 10 days post-infection, and corresponding frequencies of Siglec F⁺ alveolar macrophages, *n* = 4 mice per group shown as individual data points and mean (**e–g**). **f** gMFI of alveolar and CD11b⁺ macrophage

metabolic protein expression normalized to mean gMFI of respective population from naïve mice. **g** Representative gating for alternative activation markers and corresponding gMFI of metabolic markers segregated according to PDL2 or RELMα expression. **h** Fold change in metabolic markers across infections, relative to the matched naïve population from the same tissue. Data representative of one (Sm, *n* = 6) or two (Sm, *n* = 4/Hp, *n* = 2,5) independent experiments. Statistics calculated using one-way ANOVA with Tukey test for multiple comparisons (**d**) or two-tailed unpaired *t* test (**e**). ****$p < 0.0001$, ***$p < 0.001$, **$p < 0.01$, *$p < 0.05$. Source data are provided as a Source Data file.

flow cytometry for high dimensional, single-cell analysis of tissue macrophages, adapting 'met-flow' to interrogate cellular metabolism. Our investigation focused on mouse tissues; however, the cross-reactivity of the antibodies, as demonstrated with human MDMs, allows our panel to be readily used for human tissues as well.

Each tissue-niche has its own controlled metabolic micro-environment. Macrophages are therefore predicted to have complementary metabolic wiring to support their habitation in any particular tissue. We accordingly find diverse metabolic phenotypes in macrophages across the seven interrogated tissues. Liver-X-Receptors

(LXRs) are crucial for Kupffer Cell formation, and are known to promote glycolysis and fatty acid synthesis by activating SREBPs[19,35,36]. Accordingly, liver macrophages expressed comparably high levels of GLUT1, PKM, and ACC1. Kupffer Cells also expressed uniquely high amounts of the amino acid transporter CD98. Little is known to date on amino acid metabolism in Kupffer cells, however recent work suggests it may play a role regulating liver inflammation[37]. Overall, splenic macrophages were shown to have a markedly similar profile to Kupffer Cells, thus the transcriptional programs required in some tissues for macrophage identity may play a key role in dictating cellular metabolism, alongside the nutrient environment. Our data also align with observations that macrophages from the small intestine, despite their overall similarities to those from the large intestine, are metabolically less active[20]. Alveolar macrophages exist in a highly lipid-concentrated space and possess restricted glycolytic capacity[21], fitting with comparatively high CD36 expression and low GLUT1 expression. However, here we unexpectedly found poor expression of CPT1A in alveolar macrophages relative to other populations. One possible explanation is that surfactant is composed of complex phospholipids and sterols as opposed to free fatty acids that could be directly oxidized[38]. Interestingly, blocking oxidative phosphorylation did not alter fatty acid concentrations in alveolar macrophages, despite increased CD36 and decreased gene expression for beta-oxidation[22]. We also note that peritoneal macrophages generally possess the highest expression of metabolic proteins compared to stromal counterparts. During inflammation or damage, peritoneal macrophages are capable of rapid migration into the visceral tissue[39]. Perhaps these metabolic traits make peritoneal macrophages metabolically poised to speedily respond, migrate and adapt within a new tissue-destination. Remarkably, peritoneal macrophages are able to almost completely adopt the transcriptional identity of alveolar macrophages when transferred into the lung, including upregulation of metabolic transcription factors[24].

Elegant work from Levin et al. has previously identified the tissue environment as a key determinant of macrophage identity at the transcriptional level, of which metabolic pathways were frequently a defining characteristic[24]. Comparing our data with theirs, we found overall a similarity between the mRNA and protein level of the targets assessed. CD36 and GLUT1, which showed the greatest discordance, are also both regulated by endocytosis and lysosomal degradation[28,40], providing one possible justification for the differences. However, this comparison also underlines the need to consider the dissociation between transcription and translation. A poignant example is the low correlation between the transcriptome and proteome of activated T cells, in particular for glycolytic proteins[41,42].

Monocytes and macrophages have been well-studied for their metabolic characteristics in isolation, but the metabolic underpinnings of adult monocyte-to-macrophage differentiation have yet to be fully defined. Several pieces of evidence strongly support the event of a metabolic switch during this transition[43–45]. Accordingly, we found notable differences between monocytes, early monocyte-derived macrophages and phenotypically mature macrophages. Most strikingly in the peritoneal cavity, immature MHCII[Hi] macrophages displayed elevated metabolic enzyme expression involved in glycolysis and fatty-acid oxidation, which was not evident in F4/80[Hi] cells. It has been shown previously that a key regulator of lipid metabolism, PPARγ, is elevated in thioglycolate-elicited intermediate macrophages relative to Ly6C[Hi] monocytes, and deficiency prevents normal differentiation into resident macrophages[45]. It was more recently demonstrated that CCR2-controlled deletion of CPT1A abrogates differentiation of monocyte-derived macrophages following cardiac transplant[46]. Together with our data, these studies indicate a rapid shift, and requirement, for glycolysis and fatty-acid oxidation in the process of macrophage development from monocytes in the peritoneal cavity. Interestingly, lipid metabolism appears tightly regulated in resident peritoneal macrophages. The lipid receptor/transporter ApoE

is a key feature of TIM4[+] macrophages[5], and here we show increased ACC1 expression in these cells. As a central role of resident macrophages is the clearance of dead cells via efferocytosis, it is tempting to speculate that lipid accumulation may be required to facilitate membrane organization and organelle synthesis to support this process. As evidence for this hypothesis, ACC1 expression correlated with efferocytotic capacity, and blocking ACC activity impaired apoptotic cell clearance.

One of our most salient findings was the diverse metabolic states of TIM4[+] macrophages in the intestines. TIM4[+] macrophages in the colon have been described as embryonically seeded[8], and are frequently treated as a homogenous population. We show in both intestines, this is not the case, as CD11c and CD206 can be used to identify two metabolically distinct subsets that closely mirror the TIM4[–] equivalents. Interestingly, TIM4[+] macrophage are more likely to arise from monocytes in the small versus large intestine[8], which could indicate shortened survival and increased turnover of resident macrophages in the small intestine. Accordingly, there are significantly fewer TIM4[+] cells in the small intestine, and those present have significantly reduced mitochondrial fitness compared to both monocytes and colonic macrophages. These differences could be partially regulated by the increased abundance of microbial commensals in the large bowel, that produce short-chain fatty acids capable of sustaining mitochondrial metabolism in lamina propria macrophages[47]. We also observed prominent increases in PDL1 expression in small intestinal macrophages, largely in CD11c[+] macrophages. Interestingly, IFNγ is a shared trigger for cell death in macrophages and PDL1 expression[48,49], and can be produced spontaneously by intraepithelial lymphocytes that are present at 10–20x the numbers seen in the large intestine[50,51]. Thus inflammatory signaling, contrasted with tolerogenic signals in the large intestine, may provide an explanation for the overall reduction in metabolic fitness of the small bowel macrophages and represents an interesting hypothesis for further investigation. The exact link between PDL1 and metabolism in the intestine also remains a point of future interest. There are several examples of PDL1 both controlling, and being controlled by cellular metabolism[52,53]; it will therefore be interesting to pursue whether decreased ACC1 expression could be a cause or consequence of increased PDL1 in intestinal macrophages.

We further wanted to determine how the metabolism of tissue macrophages is augmented following infection. To this end we used three different helminth infection models to examine whether alternative activation of macrophages in different organs leads to metabolic changes consistent with in vitro stimulation. In all cases, evidence for increased fatty acid synthesis/oxidation, glycolysis and amino acid uptake was witnessed in PDL2[+] and/or RELMα[+] macrophages from infected mice, in agreement with the important roles previously shown for these pathways in alternative activation of macrophages[15,30,31]. Even so, our data suggest that changes in these pathways may be an additional consequence of their monocytic origin. Indeed, focused analysis of intestinal resident macrophages suggested they are particularly hyporesponsive, consistent with ex vivo responses to LPS[32]. Nevertheless, persisting alveolar macrophages had a strong metabolic response to Nb infection. This was unexpected in light of recent work from Svedberg et al. showing, at earlier time points, alveolar macrophages fail to express RELMα and the authors attribute this to inadequate glycolytic capacity[21]. Our results may reflect more coordinate timing with the peak of the adaptive Th2 response[54], however as direct administration of IL-4 was insufficient to stimulate alveolar macrophages[21], additional immune/metabolic signals are likely present in the context of Nb infection that support metabolic reprogramming. Together, our data highlight that while metabolic rewiring of alternatively activated macrophages to helminth infection across tissues are largely conserved, which likely reflects conserved responses to the type 2 cytokine milieu, there are also clear differences that may

indicate tissue-enforced programming, nutrient status, or differential states of maturation.

Overall, despite the well-recognized importance and heterogeneity of tissue macrophages, a comprehensive picture of their metabolic diversity is still lacking. With our work combining spectral-flow cytometry and metabolic protein analysis, we have provided a foundation for continued exploration of the in vivo aspects of macrophage metabolism.

## Methods

All research is compliant with the relevant ethical bodies. Animal experiments were complete in compliance with the Guide for the Care and Use of Laboratory Animal Research, and with approval from the Dutch Central Authority for Scientific Procedures on Animals (CCD; license number: AVD1160020198846).

### Mice and infections

C57BL/6 mice were used, bred in-house and maintained under specific pathogen free (SPF) conditions, or purchased from Envigo (C57BL/6JOlaHsd, 057). Mice were used between 8–20 weeks of age. *H. polygyrus* infection was given by oral gavage of 200 L3 larvae. Infections were cleared by oral gavage of 2 mg (200ul in ddH2O) pyrantel pamoate on day 14 post-infection. Effective infection and treatment was confirmed by fecal egg counts. *N. brasiliensis* infection was achieved by sub-cutaneous injection of 250L3 larvae. *S. mansoni* infection was done by percutaneous exposure of 35 cercariae as described previously.

### Mouse bone marrow-derived macrophages

Femurs and tibias of mice were collected in RPMI (ThermoFisher, 61870036), surface sterilized with ethanol and flushed with cold HBSS (ThermoFisher, 24020117) using a 25 g needle and syringe. Clumps and fragments were removed by aspirating up and down with the syringe followed by filtering through a 40 µM strainer into a 50 ml tube. Bone marrow cells were centrifuged at $300g$ for 5 min, resuspended and counted in cold RPMI. Cells were adjusted to concentration in RPMI containing 10% heat-inactivated FBS (Serana, S-FBS-SA-015), 100U/ml penicillin (Eureco-pharma), 100 µg/ml streptomycin (Sigma, S9137), 50 µM 2-mercaptoethanol (Sigma, M3148) and 10% supernatant of M-CSF producing L929 cells. 5 ml containing $2 \times 10^6$ cells were plated in non-culture treated 6-well plates. Cultures were supplemented on day 3 or 4 with 5 ml of complete RPMI containing 20% L929 supernatant. Cells were harvested using accutase (Sigma, A6964) on day 7, replated in 24-well plates with $(3–5) \times 10^5$ cells/well and stimulated in 1 ml for 18–24 h with LPS (100 ng/ml) and IFNγ (50 ng/ml, BioLegend, 575304) or IL-4 (20 ng/ml, BioLegend, 574304).

### Human monocyte-derived macrophages

Monocytes were isolated from buffy coats of healthy volunteers donated anonymously to the national blood bank Sanquin (Amsterdam, Netherlands) with informed consent. Blood was diluted 1:3 in HBSS and placed on 12 mL of Ficoll to obtain mononuclear cells. The solution was centrifuged at $400g$ for 30 min at RT, without brake. The resulting mononuclear cell layer was removed, placed in a new tube containing HBSS, and centrifuged at $300g$ for 20 min. The cells were washed twice more and monocytes were isolated using CD14 MACS beads (Miltenyi, 130-118-906) according to the manufacturer's recommendations, routinely resulting in a monocyte purity of >95%. To differentiate into macrophages $2 \times 10^6$/well of monocytes were cultured in 6 well plates using RPMI (ThermoFisher, 21870076) supplemented with 10% FBS, 100 U/mL of penicillin, 100 µg/mL streptomycin, 2 mM of glutamine, and either 20 ng/mL rGM-CSF (ThermoFisher, PHC2011) or 20 ng/ml rM-CSF (ThermoFisher, PHC9501) for 6 days. Cultures were supplemented on day 2/3 with an equal volume of 2x concentrated cytokines.

### Seahorse metabolic analysis

$1 \times 10^5$ mouse BMDM were plated for stimulation in an XFe96 well Seahorse plate (Agilent) and stimulated overnight. Media was replaced, after 2x washes with PBS, with 80 µl XF assay made from base RPMI (Sigma, R6504) supplemented with 10 mM glucose (Sigma, G8644), 2 mM glutamax (ThermoFisher, 35050061), and 2 mM sodium-pyruvate (ThermoFisher, 11360070), and incubated in a non-$CO_2$ 37 °C incubator for 1 h. Immediately before analysis, an additional 95 µl of XF media was added to all cells. As cells were incubating injected compounds were diluted in XF media and added to the hydrated cartridge, after which the cartridge was immediately loaded into the Seahorse for calibration. Oligomycin = 2 µM (Cayman, 11342), FCCP = 1.5 µM (Sigma, C2920), Rotenone (Sigma, 557368) and Antimycin (Sigma, A8674) = 0.75 µM each (1.5 µM total).

### Tissue immune cell isolation

Peritoneal exudate was obtained by injection of 5 ml cold PBS containing 2% FBS and 2 mM EDTA (ThermoFisher, 15575020) into the exposed abdomen of sacrificed mice, which was subsequently withdrawn after ~20 s of gentle agitation, and kept on ice until counting and plating.

Published protocols were adapted for isolation of leukocytes from the small-intestine[55] and colon (including cecum)[56]. Both tissues were placed on PBS-soaked paper towel, opened longitudinally, and intestinal contents were scraped off using a metal spatula. Opened intestines were washed vigorously by shaking in a 50 ml tube containing 15 ml of Ca/Mg-free HBSS (ThermoFisher, 14170112) containing 2 mM EDTA (wash media), then cut into 1–2 cm pieces and stored in 10 ml of wash media on ice until further processing. To remove mucous and strip epithelial cells, tissues were shaken vigorously in 50 ml tubes with 10 ml pre-warmed wash media, strained through 250 µM Nitex, placed back in tubes containing 10 ml fresh wash media, and incubated for 20 min at 37 °C, shaking at 200 rpm. Small intestines were washed a total of 3 times, colons a total of 2 times. Small intestine digest: 10 ml RPMI containing 10% FBS, 1 mg/ml collagenase VIII (Sigma, C2139), 40U/ml DNAse I (Sigma, D4263) for 15–20 min (or until tissue is not quite completely digested). Colon digest: 10 ml RPMI containing 10%FBS, 1 mg/ml collagenase IV (Sigma, C5138), 0.5 mg/ml collagenase D (COLLD-RO), 1 mg/ml dispase II (Sigma, D4693), 40U/ml DNAse I for 25–30 min. Digestion was stopped by immediate addition of cold RPMI containing 10% FBS. Digested intestines were filtered through a 100µm strainer, spun at 400 g for 5 min, resuspended in FACS buffer (2% FBS, 2 mM EDTA) and filtered a second time through a 40 µM strainer, spun, resuspended and stored on ice for counting and plating.

Liver digestion was also adapted from previously published protocols[57]. Livers were taken and placed in cold RPMI, minced with scissors and immediately transferred to tubes containing 10 ml digest buffer (same as colon). Minced livers were digested for 25–30 min at 200 rpm, 37 deg, with additional manual shaking vigorously by hand every 5–8 min. Digested livers were filtered through 100um strainers, and topped-up with cold RPMI containing 10% FCS up to 50 ml, spun at 300 g for 5 min. The supernatant was removed, and cells were washed a second time with 30 ml FACS buffer. The remaining pellet was treated with ACK red-blood cell lysis buffer, washed and CD45 enriched using a MACS positive selection kit (Miltenyi, 130-052-301).

Lungs and spleens were placed in a 2 ml tube with cold PBS, removed and minced in a new 2 ml tube, before direct addition of 1.5 ml digest buffer consisting of 1 mg/ml collagenase IV and 40U/ml DNAse I and shaken for 30 min before storing on ice, mashing through 100 µM filter and washing with 10 ml cold RPMI with 10% FBS. Homogenates were red-blood cell lysed, washed, and passed through a 40µm strainer before counting and plating.

As described previously[58], brains were digested after mincing with a razor in 0.2 mg/ml Collagenase IV and 40U/ml DNAse I in a shaking

waterbath for 1 h. A 1 ml pipet was used to further dissociate digesting brains every 15 min. Homogenate was filtered through a 100uM strainer immediately before CD45⁺ MACS isolation.

## Staining procedure for spectral flow cytometry

A complete list of antibodies and dilutions is shown in Supplementary Table 1. Intracellularly stained metabolic targets were conjugated in-house using the corresponding kit according to the manufacturer's protocol. For in vitro experiments, all cells from a single well were stained after transferring to 96-well V-bottom plate. For tissues, 1-2×10⁶ cells (numbers kept consistent within experiments) were stained in V-bottom plates in 50 µl for each step. Cells were pre-stained with viability dye and Fc-block in PBS for 15 min on ice. If required, subsequent live surface-staining was performed for certain targets in FACS buffer for 30 min on ice (Supplementary Table 1) before fixation. Cells were fixed with eBioscience Foxp3 fixation/permeabilization staining kit according to the recommended protocol. Following fixation, the remaining surface targets were stained in FACS buffer for 30 min at 4 °C. All staining was done in the appropriate buffer containing 1x Brilliant Stain Buffer Plus (BD Biosciences, 566385) and TrueStain Monocyte Block (BioLegend, 426103). Cells were washed twice in perm/wash before staining intracellular targets in 1x permeabilization buffer, containing Fc-block, for 2 h at 4 deg. If staining for RELMα, primary and secondary stains were completed (30 min each) before metabolic target staining to prevent Fc-binding of the rabbit isotypes by the secondary. Cells were acquired on a Cytek Aurora 5-laser spectral flow cytometer. Acquired samples were unmixed using SpectroFlo version 3 and analyzed with FlowJo and/or OMIQ.ai.

## Spectral unmixing

See supplementary material for a point-by-point protocol

## Efferocytosis assay

For the ex vivo assay, total PEC suspension was plated in a 24-well plate (1e6 cells/well) for 2 h in serum-free media. Cells were then washed 3 times with pre-warmed media before culturing overnight in complete RPMI media with either DMSO or 60uM pan-ACC inhibitor CP 640,186 (Sanbio, 17691-5). Thymocytes were isolated in the parallel to the PEC, by dissociating the thymus in a petri plate with frosted slides, and rendered apoptotic by overnight incubation with 2 µM staurosporine (Tocris, 1285). This typically yielded ~50% Annexin V⁺7AAD⁻ cells. AC were labeled with cell-trace violet (ThermoFisher, C34557) according to the manufacturer's instructions and then 1e6 cells were added directly to cultured macrophage for 60 min. AC were removed by consecutive washes with warm PBS before placing cells on ice in FACS buffer (2% FBS, 2 mM EDTA) to be harvest by scraping. For in vitro efferocytosis, a similar protocol was followed using jurkat cells and bone-marrow-derived macrophages cultured overnight with DMSO or ACC inhibitor.

## Metabolic dye staining

After isolation, 1e6 cells from tissue single-cell suspensions were plated in a non-treated V-bottom 96 well plate. Cells were washed once with pre-warmed base RPMI and then suspended in 200ul of RPMI containing 0.5 mM BSA (Sigma, BSAV-RO), 5uM BODIPY C16 (Thermo-Fisher, D3821), 5 nM TMRM (ThermoFisher, M20036) and 20 nM MitoTracker DeepRed (ThermoFisher, M22426) for 20 min at 37 °C. Cells were washed 2x in cold PBS/2%FBS/2 mM BSA before lived/dead and surface staining and acquired immediately afterwards.

## HPG uptake (SCENITH)

5e5 peritoneal cavity cells were plated in a 96-well untreated V-bottom plate, washed with PBS and plated in 90ul of methionine-free media (Sigma, R7513) supplemented with 65 mg/L L-cystine dihydrochloride (Sigma, C6727), 1x GlutaMAX and 10% dialyzed FCS. Cells were

methionine starved for 45 min before addition of 10ul indicated inhibitor(s) (media, 2 µM Oligomycin, 100 mM 2DG [Sigma, D8375]) and subsequently incubated another 15 min. Homopropargylglycine (Click Chemistry Tools, 1067) was added at a final concentration of 100uM and incubated for 30 min before being washed 2x with cold PBS, live/dead stain and fixed with 2% PFA for 15 min.

## Click Chemistry

Cells fixed after HPG uptake were permeabilized with PBS containing 1% BSA/0.1% Saponin for 15 min and washed 2x in Click buffer (100 mM Tris-HCl, pH 7.4) before the addition of Click reaction mix. The reaction mix was made by sequential addition of 10 mM Sodium Ascorbate (Sigma, A7631), 2 mM THPTA (Click Chemistry Tools, 1010), 0.5 µM AFdye488 azide plus (Click Chemistry Tools, 1475) and 1x click buffer to CuSO₄ (0.5 mM final conc., [Sigma, 209198]). Samples were incubated for 30 min in the dark at room temperature. Cells were washed with FACS buffer and surface staining was performed as normal.

## OMIQ analysis and workflow

Populations were first gated in FlowJo and exported (Figs. 2–4, total Mph; Fig. 5, total live CD45⁺) for uploading into OMIQ. Parameters were scaled using conversion factors ranging from 6000–20000, gated according to Mph population(s) of interested, and subsampled using a maximum equal distribution across groups/tissues. After subsampling, UMAP was performed using the indicated parameters and default settings, followed by PhenoGraph clustering ($k = 100$). Data was further analyzed with EdgeR to determine significantly different clusters, or by generating heatmaps with Euclidean clustering. OMIQ was also used to generate scatter, contour and histogram plots. Graphs were generated using GraphPad Prism, showing geometric fluorescence intensity data exported from FlowJo or median fluorescent intensity as determined by OMIQ.

## Statistical analysis

Data were tested for normality using the Shapiro–Wilk test. Statistical tests used are indicated in the figure legends. Generally, data were compared using $t$ tests for two groups, one-way ANOVA for more than two groups, or two-way ANOVA for comparing multiple parameters across two or more groups, with Tukey's post-hoc test for multiple comparison. If comparing parameters within the same sample, a paired or repeated-measures test with Geisser-Greenhouse correction was used. $p$ values < 0.05 were considered significant (*$p < 0.05$, **$p < 0.01$, ***$p < 0.001$, ****$p < 0.0001$). All statistical analyses were performed using GraphPad Prism v.9.0.

## Reporting summary

Further information on research design is available in the Nature Portfolio Reporting Summary linked to this article.

## Data availability

Data for the study are available from the corresponding authors upon request. Source data are provided with this paper. Transcript data for Supplementary Figure 6e obtained from publicly available Gene Skyline Database (http://rstats.immgen.org/Skyline/skyline.html), deposited by Kim et al. (Immunological Genome Project). Sequencing data used in Supplementary Fig. 5 was obtained from manuscript source data from Lavin et al. [24]. Source data are provided with this paper.

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

## Acknowledgements
This work was supported by the LUMC, by the Leiden University Fund/Schild-de Groen Fonds, (W213032-2-38, www.luf.nl, W213032-2-38) awarded to G.A.H, and by an NWO Vidi grant (91719349) awarded to B.E. T.T. was supported by funding from the LSH-TKI (DC4Balance, LSHM18056-SGF). R.M.M. is funded by a Wellcome Trust Investigator Award (219530). We would like to acknowledge LUMC Flow Core Facility operators, in particular IJsbrand Reyneveld, for the continual maintenance and trouble-shooting of the Cytek Auroras. Many thanks to our colleagues in LUMC Parasitology for their continual scientific discussions and critical reading of the work.

## Author contributions
G.A.H. and B.E. conceived the study. G.A.H. designed, performed and analyzed experiments with the assistance of T.A.P. and L.A. T.T. aided with analysis. G.P.W provided cells from N.b. infected lungs. F.V. and R.S. developed the click-based SCENITH assay and provided reagents. R.M.M. provided the H.p. and N.b larvae. G.A.H. and B.E. drafted and edited the manuscript.

## Competing interests
The authors declare no competing interests.
