## [Peer Review File · Nature Communications]

Metabolic heterogeneity of tissue macrophages in steady-state and helminth infectionReviewers' comments:

Reviewer #1 (Remarks to the Author):

In this article, the authors use spectral cytometry to demonstrate metabolic differences in peritoneal and intestinal macrophages at single-cell resolution in steady state and in response to infection. Especially from a technological perspective, this could be an important article for the community. Yet, for this purpose the material and methods section should be extended and details should be provided to allow researchers to perform this kind of analysis.

Scientifically, the article is less exciting as it is only descriptive and new insights are rather limited (which is not an issue for a "methods" paper).

In any case, there are some points that need to be addressed before being suitable for publication.

- It should be verified that the observed metabolic differences correspond to earlier described expression and metabolic profiles (eg Lavin et al, 2014, Cell). In relation to this, other tissue macrophages (spleen, liver, brain,...) could be profiled to demonstrate that the obtained profiles are not just a lucky shot.
- Human validation is important but the M-CSF vs GM-CSF comparison is too limited and should be extended by including the same conditions as performed in the mouse studies. Including an LPS only condition would also be important as this would allow the comparison with an earlier cyTOF-based single-cell metabolic analysis of macrophages (Hartmann et al, Nat Biotechnology)
- The authors performed phenograph clustering which is an elegant and unbiased way to analyze multiplex data. Yet, they later further subdivide the obtained cluster 10. This biased way of further splitting should be better clarified as it would make more sense to compare the obtained clusters.
- Importantly, the obtained data should be validated using established techniques and particularly western blot analysis to show that the obtained signal by flow are "real" and vice versa, that no changes are missed. This is probably not possible on tissue macrophages but could be done using differentially treated BMDM or other suitable positive and negative

controls.

- Prevent claims of being the first to apply this technique as other already did this using cyTOF and actually also using Cytex Aurora on myeloid cells (eg Thompson, ... Powell, Cell Reports, 2021)

- Fig 1d: show absolute MFI and not relative MFI

- Fig 5i: this way of showing the data is difficult and not very informative for the reader. Not sure how to solve this, but wanted to mention that this is a difficult presentation style to follow

Reviewer #2 (Remarks to the Author):

This study by Heieis et al is a beautiful report describing a new approach to characterizing macrophage metabolism in vivo. The authors carefully and thoroughly detail the profile of metabolic proteins in macrophages from different tissues and with and without intestinal parasite infection using spectral flow cytometry. The novelty of the work is the technique and the profiling of macrophages in vivo, rather than the typical metabolic profiling of in vitro treated macrophages. This report will be a valuable tool for other groups that wish to examine macrophage metabolism and function in vivo. The methods are well described and the main text of the manuscript is clearly written.

The drawback of the manuscript is that it is largely descriptive and is more a methodological report. The only perturbation the authors make to their system is to infect mice with *H. polygyrus* and characterize large peritoneal macrophages and small intestinal macrophages with infection compared with uninfected cells. While this is valuable information and confirms that their technique can detect changes to macrophages in vivo, the data itself does not necessarily significantly advance our knowledge or address a hypothesis. Questions that could be addressed are whether metabolic changes are sustained during helminth infection; if the metabolic changes remain even if the helminth infection is cleared; how the metabolic changes with helminth compare with a bacterial or viral challenge.

Specific comments:

1. Lines 55-60: Be more specific in the text about the changes you see. What is upregulated and downregulated?

2. Figure 1 e-f: What do the dots represent? Individual donors?
3. Sup. Fig. 2b: Make the first panel bigger. It's very hard to see the different populations even with the figure blown up.

Reviewer #3 (Remarks to the Author):

The manuscript by Heieis et al., describes a method by which the ex vivo metabolic exploration of tissue resident macrophages can be mapped. The authors utilise a previously published 'Met-Flow' strategy in order to investigate the metabolic requirements of different tissue macrophage populations both at steady state and post-helminth infection. Overall, the manuscript is interesting, but largely descriptive based on the abundance of a panel of metabolic markers. I have a few points to consider:

Major:

1. Most of the findings presented were reliant solely on abundance of metabolic proteins/enzymes but may not relate to their activity. Do the authors have additional data/evidence that abundance translates into activity for example, measuring fatty acid uptake and CD36 expression, mitochondrial mass/membrane potential etc. Furthermore, the authors should explore the recently developed SCENITH assay with corresponding inhibitors for the various proteins measured outside the traditional oligomycin/2-DG employed in the original paper to support their findings.
2. It is stated that post-infection SLIP mphs decrease levels of SDHA whilst increasing GLUT1, can the authors confirm if this reduction of SDHA is indicative of a global reduction in mitochondrial mass, or whether it is SDHA specific.

Minor:

1. Why were the Seahorse assays in Figure 1 conducted differently? i.e. the human MDMs were presumably glucose starved then introduced to a 10 mM glucose injection, whereas the mouse BMDM were not.
2. Can the authors explain the discrepancies between the mouse BMDM IL-4 or LPS/IFN γ ECAR graph and a recently published manuscript entitled 'An integrated toolbox to profile macrophage immunometabolism'. Furthermore, given this paper highlights the

investigation of macrophage metabolism at the single cell level it should be cited within the introduction as such methodologies could be extended to what is proposed in the current paper.

Thank you very much to the reviewers for contributing their time to the reviewing of our manuscript, and for both the positive feedback and insightful critique/suggestions.

In general, it is clear that we had written the manuscript in a way that read much more as a methodological report which was not our intent, as others have published met-flow based techniques prior to us. Yet we understand how the original draft could be seen as such. The motivation to highlight the use of spectral flow was in part due its relative novelty/advantages, but mainly to aid the flow community in overcoming the challenges that we faced in switching to this technology. Our main intent for the paper was to demonstrate that metabolic phenotyping of macrophages from tissues is possible, as reliance on in vitro BMDM is unfortunately still the norm, and (as far as we have seen) met-flow has yet to be applied to tissue macrophages and represents an important next step in the field. Due to the way we had drafted the previous version it is also evident that we did not sufficiently emphasize our novel findings from employing this strategy. We hope with the revised version, and the additional data/analysis, we have better accomplished this.

Provided below is a point-by-point response (in blue) to the other specific concerns raised.

Reviewer #1

In this article, the authors use spectral cytometry to demonstrate metabolic differences in peritoneal and intestinal macrophages at single-cell resolution in steady state and in response to infection. Especially from a technological perspective, this could be an important article for the community. Yet, for this purpose the material and methods section should be extended and details should be provided to allow researchers to perform this kind of analysis.

Scientifically, the article is less exciting as it is only descriptive and new insights are rather limited (which is not an issue for a "methods" paper. In any case, there are some points that need to be addressed before being suitable for publication.

- It should be verified that the observed metabolic differences correspond to earlier described expression and metabolic profiles (eg Lavin et al, 2014, Cell). In relation to this, other tissue macrophages (spleen, liver, brain,...) could be profiled to demonstrate that the obtained profiles are not just a lucky shot.

The cited paper is indeed very relevant and interesting, and we have further included the spleen and brain in our analysis (Fig. 2). As we have now performed numerous repeats of the analysis, and have also observed many differences supported by other studies (i.e. high metabolic phenotype of Lg. Int compared to Sm. Int), we are confident in the reliability of the data. At the reviewers suggestion we compared the RNAseq data from Lavin et al to our met-flow data and added this analysis to the supplementary data (Sup Fig. 5). While some targets show a clear match in protein and mRNA, others do not. We believe there are several possible reasons for this, which we have tried to address further in the discussion (paragraph starting at line 350).

- Human validation is important but the M-CSF vs GM-CSF comparison is too limited and should be extended by including the same conditions as performed in the mouse studies. Including an LPS only condition would also be important as this would allow the comparison with an earlier cyTOF-based single-cell metabolic analysis of macrophages (Hartmann et al, Nat Biotechnology)

Our reason for choosing an M-CSF vs GM-CSF comparison is due to the strong relationship between the respective cytokines and regulatory or inflammatory phenotypes, and that they are known to drive significantly divergent metabolic programs which better allow us to validate the panel (PMIDs:

17082649, 21240265, 15070757, 34740864). Nonetheless as suggested, we performed the same staining on MCSF hMDM stimulated with LPS, LPS/IFN γ , or IL-4. Interestingly there were evident changes in metabolic capacity by Seahorse, but compared to GM-CSF, changes in metabolic protein expression were relatively minor (see figure below). Certain expected changes were still observed (increased ACC1 and CD98 in IL-4, decreased CD36 for LPS+/-IFN γ), but other changes that were observed by Hartmann et al were not obviously visible such as in GLUT1 and SDHA expression. Our different observations for GLUT1 expression can be explained by different staining protocols as Hartmann et al did a surface stain whereas we stained intracellularly due to the known internalization/recycling of the transporter – thus the significant increase seen by Hartmann et al is possibly due to surface shuttling rather than differential expression. For example, we see that only 15min of culture with 2DG causes rapid surface detection of GLUT1 on peritoneal macrophages that is not detected in the control cells, but this also blocks translation (Fig. 3d) suggesting the expression is most likely not from de novo protein synthesis. Additionally inter-donor variability is a well-recognized complication in studying hMDM, and it is clear that 1 of 2 donors in the Hartmann paper did not show a significant SDHA increase. Other obvious differences may also contribute, such as the use of IMDM vs RPMI and different serum batches/concentrations. We have decided to leave this additional data out of the manuscript as it is ultimately only meant as a prelude to the in vivo data, and to demonstrate that the use of the panel is possible with human samples; we hope these things are sufficiently addressed with the data presented.

- The authors performed phenograph clustering which is an elegant and unbiased way to analyze multiplex data. Yet, they later further subdivide the obtained cluster 10. This biased way of further splitting should be better clarified as it would make more sense to compare the obtained clusters.

We have redone the analysis by making minor changes in the scaling to better define the CD11c+ population. The resulting clusters of the new analysis provide a much better match to the description of peritoneal macrophages, eliminating the need for manual separation (Now supplementary Fig. 6a-c).

- Importantly, the obtained data should be validated using established techniques and particularly western blot analysis to show that the obtained signal by flow are "real" and vice versa, that no changes are missed. This is probably not possible on tissue macrophages but could be done using differentially treated BMDM or other suitable positive and negative controls.

While we agree that other techniques are important for validation, the antibodies used have been extensively validated by the manufacturers, and been published for the use of flow-based techniques previously (PMID: 32533056, 32868913, 34965415, 33705706). We have now, however, provided several new pieces of data that add confidence to results obtained through spectral-flow. First, for BMDM we have qPCR data showing overlap between gene and protein expression (see figure below, and which we could add to the manuscript if preferred). Secondly, as per the reviewers earlier comment, we have compared RNA seq data for tissue macrophages from Lavin et al to our met-flow data showing an overall similarity. Finally, we also direct the reviewer to our response to the first two comments by Reviewer #3 in which we compared mitochondrial staining/fatty acid uptake to metabolic protein expression, and successfully correlated protein expression of glycolytic and oxphos enzymes with reliance on the processes for ATP synthesis via SCENITH (Fig. 3d&e, Fig.4c).

- Prevent claims of being the first to apply this technique as other already did this using cyTOF and actually also using Cytek Aurora on myeloid cells (eg Thompson, ... Powell, Cell Reports, 2021)

We agree that it is best to remove these statements as a precaution, and we have done so.

- Fig 1d: show absolute MFI and not relative MFI

The main reason to show relative MFI is to permit pooling of samples from repeat experiments, which we believe strengthens the data. However we have now tried to include either representative histograms and/or graphs with the absolute MFI for all experiments throughout the manuscript, either in the main or supplemental, to ensure raw data is also visible.

- Fig 5i: this way of showing the data is difficult and not very informative for the reader. Not sure how to solve this, but wanted to mention that this is a difficult presentation style to follow

Thanks very much for mentioning. We agree it was not the best visualization and want to ensure data is easy to interpret. The panel is no longer in the manuscript due to the significant changes/additions, but we hope that the new data are presented in much clearer way.

Reviewer #2

This study by Heieis et al is a beautiful report describing a new approach to characterizing macrophage metabolism *in vivo*. The authors carefully and thoroughly detail the profile of metabolic proteins in macrophages from different tissues and with and without intestinal parasite infection using spectral flow cytometry. The novelty of the work is the technique and the profiling of macrophages *in vivo*, rather than the typical metabolic profiling of *in vitro* treated macrophages. This report will be a valuable tool for other groups that wish to examine macrophage metabolism and function *in vivo*. The methods are well described and the main text of the manuscript is clearly written.

The drawback of the manuscript is that it is largely descriptive and is more a methodological report. The only perturbation the authors make to their system is to infect mice with *H. polygyrus* and characterize large peritoneal macrophages and small intestinal macrophages with infection compared with uninfected cells. While this is valuable information and confirms that their technique can detect changes to macrophages *in vivo*, the data itself does not necessarily significantly advance our knowledge or address a hypothesis. Questions that could be addressed are whether metabolic changes are sustained during helminth infection; if the metabolic changes remain even if the helminth infection is cleared; how the metabolic changes with helminth compare with a bacterial or viral challenge.

Thanks very much for the positive comments. We acknowledge the largely descriptive nature of the work. To address this point of critique we have now provided some mechanistic data, by interrogating the functional link between the strong correlation we observed between ACC1 expression and efferocytotic capacity in macrophages from different tissues. As fatty-acid synthesis has been linked to phagocytotic processes we hypothesized that efferocytosis may similarly require increased fatty-acid synthesis to support membrane expansion for the engulfment of apoptotic cells. Accordingly, peritoneal macrophages efficiently phagocytosed dying cells *ex vivo*, whereas alveolar macrophages lacking TIM4 and possessing low ACC1 expression, did not take up apoptotic cells in the same timeframe (Fig. 2f,g). Moreover, pharmacological inhibition of ACC activity blocked efferocytosis by peritoneal macrophages, as well as *in vitro* by BMDM (Fig. 3h). Overall these data support the use of determining metabolic protein expression to identify mechanistic links between into tissue-macrophage metabolism and function.

In addition, as suggested by the reviewer, we have added new pieces of data to strengthen the novelty of the manuscript, and we have also tried to better highlight the novel findings that were possibly not sufficiently emphasized in the previous version of the paper. This includes the metabolic hypo-responsiveness of intestinal macrophage in *Hp* infection compared to the rapid influx of alternatively activated monocyte-derived macrophages that are metabolically highly active (Fig. 5d-j) as well as the presence of multiple TIM4+ resident macrophages in the intestine with different metabolic phenotypes (Fig. 4d-h). As suggested, we have also analyzed the metabolic phenotype of macrophages 5 weeks after clearance of the parasite, and found remarkable similarity to naïve mice

(Fig. 5k-n). Finally, additional helminth infections were also performed using different parasites with different tissue tropisms for their lifecycle (Fig. 6). Generally this revealed an overall similarity in the markers that are changed, suggesting a common metabolic phenotype across AAM that aligns with previous studies. However, there were also notable differences in the degree of change depending on the tissue. As the majority of AAM have a phenotype resembling monocyte-derived cells, these differences may reflect the difference between tissues macrophages at steady-state (the high CD98 expression in KC for instance).

Specific comments:

1. Lines 55-60: Be more specific in the text about the changes you see. What is upregulated and downregulated?

Due to the revisions this text has been removed. Where possible we have tried to be specific about changes observed that are of particular interest, but did not point out all differences in an effort to keep the manuscript as succinct/readable as possible.

2. Figure 1 e-f: What do the dots represent? Individual donors?

All datapoints represent individual mice/donors. We have clarified this in the text.

3. Sup. Fig. 2b: Make the first panel bigger. It's very hard to see the different populations even with the figure blown up.

As a result of the extensive revisions this figure has been removed.

Reviewer #3

The manuscript by Heieis et al., describes a method by which the ex vivo metabolic exploration of tissue resident macrophages can be mapped. The authors utilise a previously published 'Met-Flow' strategy in order to investigate the metabolic requirements of different tissue macrophage populations both at steady state and post-helminth infection. Overall, the manuscript is interesting, but largely descriptive based on the abundance of a panel of metabolic markers. I have a few points to consider:

Major:

1. Most of the findings presented were reliant solely on abundance of metabolic proteins/enzymes but may not relate to their activity. Do the authors have additional data/evidence that abundance translates into activity for example, measuring fatty acid uptake and CD36 expression, mitochondrial mass/membrane potential etc.

We thank the reviewer for bringing up this valid point. In addition to the met-flow data, we have now included analysis using Bodipy C16, TMRM and mitotracker to further assess mitochondrial dynamics and fatty acid uptake as suggested. In regards to fatty-acid uptake, we interestingly find that CD36 is not necessarily a good predictor of uptake in the case of peritoneal macrophages (fig. 3), it is unclear whether this is due to endocytosis of CD36 as had been reported (PMIDs: 12947091, 32958780), or an alternative mechanism of uptake, such as pinocytosis. Unfortunately CD36 staining does not work after fixing the cells, so we were unable to fully address this, but public databases show high CD36 gene expression in the MHCII+ macrophages and we do observe significant changes in detection after short incubation at 37deg (Sup Fig. 6e,f). In the case of the intestine, CD36 and C16

uptake correlated much better when comparing monocytes vs MHCII⁺ macrophages (Fig. 4a & e). Mitochondrial mass and membrane potential measurements overall paralleled SDHA/Cytc staining, as seen when comparing MHCII vs F480 macrophages (Fig. 3e), as well as monocytes vs macrophages in the large intestine (Fig. 4c). Interestingly, macrophages from the small intestine displayed selectively reduced mitochondrial mass and membrane potential (Fig. 4b) which we speculate to reflect cell death rapidly after maturation, based on several pieces of literature pointing towards impaired longevity of macrophages in the small intestine. This is now discussed in the paragraph starting at line 377.

Furthermore, the authors should explore the recently developed SCENITH assay with corresponding inhibitors for the various proteins measured outside the traditional oligomycin/2-DG employed in the original paper to support their findings.

As per the reviewer's suggestion, we applied this method to the peritoneal macrophages to see whether differences seen in our panel data were also associated with altered translational dependencies (Fig. 3d). We found indeed that the increase in multiple catabolic pathway markers in MHCII⁺ macrophages correlated with more metabolic flexibility in supporting translation, whereas F480⁺ macrophages relied much more heavily on glycolysis.

To use a new set of inhibitors in SCENITH is an interesting prospect. However, as the readout of SCENITH is directly coupled to production of ATP, this technique is not suitable for exploring new inhibitors of pathways that do not directly affect ATP production. Therefore it would be hard to interpret data if we were to target such pathways.

2. It is stated that post-infection SLIP mphs decrease levels of SDHA whilst increasing GLUT1, can the authors confirm if this reduction of SDHA is indicative of a global reduction in mitochondrial mass, or whether it is SDHA specific.

After repeating the experiments the reduction in SDHA was not consistent and pooling data suggested there was minimal change on the whole population level, but a significant increase was observed in differentiating monocytes and AAM in infection (Fig. 5h). The latter observation was paralleled by increases in both MitoTracker and TMRM staining in specifically PDL2⁺ cells (supplemental Fig. 8a), indicating an overall increase in mitochondrial mass. This was also supported by minor (but not significant) increases in Cytc/ATP5A (Fig. 5h, n). Consistent with the hyporesponsive phenotype of resident macrophages no changes were observed in TIM4⁺ cells for mitochondrial staining (Supplementary Fig. 8a).

Minor:

1. Why were the Seahorse assays in Figure 1 conducted differently? i.e. the human MDMs were presumably glucose starved then introduced to a 10 mM glucose injection, whereas the mouse BMDM were not.

As suggested by the reviewer we have corrected this inconsistency by repeating the Seahorse assays with identical experimental setup for both human and murine Mphs (Supplementary Fig. 1f). This did not impact the overall differences observed.

2. Can the authors explain the discrepancies between the mouse BMDM IL-4 or LPS/IFN γ ECAR graph and a recently published manuscript entitled 'An integrated toolbox to profile macrophage

immunometabolism'. Furthermore, given this paper highlights the investigation of macrophage metabolism at the single cell level it should be cited within the introduction as such methodologies could be extended to what is proposed in the current paper.

The results we show are typical of LPS/IFN γ stimulated macrophages, in which the high ECAR at baseline with glucose is supported by the obvious yellowing of the media (before changing into assay media). Similarly, the lack of OCR is supported by the high iNOS expression. The differences may therefore be due to the starvation step introduced by Verberk et al in their paper, however the maximal ECAR are still quite comparable between our results and theirs, suggesting the macrophages are unable to restore glycolysis to basal levels following starvation. We have now cited this work in the introduction (line 58).

REVIEWERS' COMMENTS

Reviewer #1 (Remarks to the Author):

The authors did a great job in rewriting the manuscript and performing additional experiments as suggested by myself and the other reviewer. I don't have further comments and would support the publication of this article if also the other reviewer and the editor are satisfied with this revision.

Reviewer #2 (Remarks to the Author):

The authors have significantly re-worked the manuscript and included substantial new data. They have also made it less of a Methods paper and more of a Research paper. They added experiments showing that curing *H. polygyrus* infection led to a reversion of intestinal macrophage phenotype to a metabolic state similar to pre-challenge. They also added infections with other parasites that colonize different tissues to compare metabolic phenotypes. While it is still very descriptive, it does begin to offer new insights into the metabolic programming in diverse populations of macrophages, across tissues and will different infections. I think this will be of interest to the scientific community and will provide a foundation for future hypothesis-driven questions.

Reviewer #3 (Remarks to the Author):

The authors have addressed my concerns in full. In this Reviewer's opinion the manuscript is much improved and I fully recommend publication. Congratulations to the authors for an impressive study.

We would like to thank the reviewers very much for their time and pushing us to create a greatly improved manuscript. We are very happy with the final outcome and excited to share our work with the wider community.

Sincerely on behalf of all authors,

Graham Heieis & Bart Everts